# Test-Time Domain Generalization for Image Super-resolution

**Zai-Zuo Tang, Yu-Bin Yang** [*]
State Key Laboratory of Novel Software Technology
Nanjing University
Nanjing, China
`tangzz@smail.nju.edu.cn, yangyubin@nju.edu.cn`

## Abstract

Test-time domain generalization (TTDG) methods enhance the performance of neural networks on target domains by transferring the feature distribution of target samples to approximate that of the source domain, while avoiding the computational cost associated with fine-tuning on the target domain. However, existing TTDG methods primarily rely on style transfer strategies operating at a coarse granularity, which prove ineffective for pixel-level prediction tasks such as image super-resolution (SR). To address this limitation, we propose a multi-codebook based test-time domain generalization framework (MC-TTDG). Our method leverages both domain-specific and domain-invariant codebooks to achieve fine-grained representation learning on source domains, and performs pixel-level nearest-neighbor feature matching and transfer to accurately adjust target domain features. Furthermore, we introduce a voting-based strategy for optimal domain-specific codebook selection, which improves the precision of feature transfer through multi-party consensus. Extensive experiments across diverse data distributions, and network architectures demonstrate that the proposed method effectively transfers feature distributions for SR networks. Our code is available at https://github.com/ZaizuoTang/MC-TTDG.

## 1 Introduction

In real-world scenarios, due to variations in imaging conditions, the distribution of the test sample set (target domain) often diverges from that of the training set (source domain) used for neural networks (Guo et al., 2025; Deng et al., 2025). This discrepancy, commonly referred to as domain shift, can significantly degrade the predictive performance of neural networks trained on the source domain when applied to the target domain (Wang et al., 2025; Choe et al., 2025).

As illustrated in Figure 1, conventional domain generalization methods (Lee et al., 2025; Rathore et al., 2025) utilize samples from multiple source domains with divergent distributions as input. By learning domain-invariant representations across these source domains, they enhance the model's robustness on target domains with differing distributions. However, these conventional methods neglect the distribution of the target domain during testing, resulting in underutilization of target domain information. In contrast, test-time domain generalization methods (TTDG) (Ma et al., 2024; Meng et al., 2025; Nam & Lee, 2025) preserve the distributional centroids of the source domains during training. During testing, they perform style transfer on target domain samples to align their distribution with these source domain centroids. By transferring the distribution of target domain samples in this manner, TTDG methods eliminate the need for computationally expensive fine-tuning on the target domain while simultaneously improving the model's performance on target domain samples.

Low-level vision tasks, such as image super-resolution (SR), are widely employed in remote sensing imaging, image transmission, and even serve as preprocessing steps for downstream tasks. However, due to discrepancies in imaging devices, domain shift also exists in low-level vision tasks, which

---

[*]Corresponding author

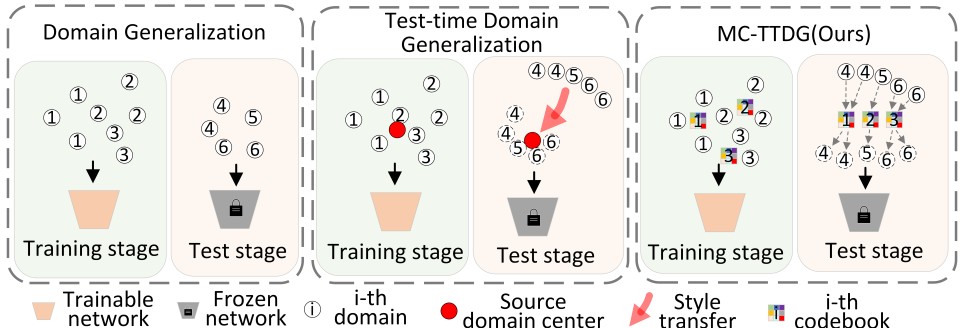

Figure 1: Comparison of domain generalization methods. The domains 1 to 3 represent the source domains, while domains 4 to 6 represent the target domains. Conventional domain generalization methods optimize the source-domain network to enhance its robustness. In contrast, TTDG methods optimize the target-domain samples by aligning their distribution with the source-domain distribution, thereby improving the performance of models trained on the source domain when applied to target data. Existing TTDG methods typically preserve the source distribution using a single centroid and perform adjustment via style transfer (As shown in Equation 1). The proposed method (MC-TTDG) improves upon this framework by introducing domain-specific codebooks to retain domain-specific features and enabling pixel-level highly fine-grained feature transfer.

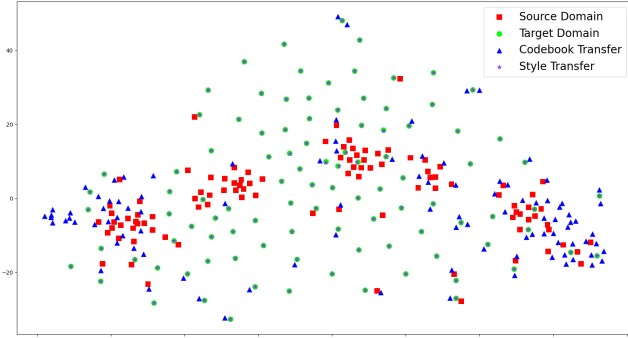

Figure 2: Comparison of feature transfer methods. We compare existing style transfer and the proposed codebook-based transfer in aligning the target domain distribution with the source domain distribution. Due to its inability to perform fine-grained adjustments at the pixel level, style transfer proves highly ineffective for low-level vision tasks such as SR, resulting in most transformed samples overlapping significantly with the original target domain samples. In contrast, the proposed codebook transfer method effectively aligns the target domain distribution with the source domain. (t-SNE (Maaten & Hinton, 2008) is used for feature dimensionality reduction.)

hinders the performance of deep neural networks in these applications (Tang & Yang, 2024; Ai et al., 2024).

However, applying existing TTDG methods to SR tasks faces the following three challenges:

**Challenge1: Low granularity in feature transfer.** Existing test-time domain generalization methods utilize style transfer (Huang & Belongie, 2017) (as shown in Equation 1) to modify the mean and variance of target domain features, thereby approximating the feature distribution of the source domain. Such global feature transfer is well-suited for high-level vision tasks like image classification, which rely on abstract representations of entire images. However, as shown in Figure 2, for pixel-level tasks such as image SR, style transfer fails to perform fine-grained adjustments for each individual pixel. Consequently, it is difficult to accurately align the target domain feature distribution with the source domain feature distribution in SR tasks.

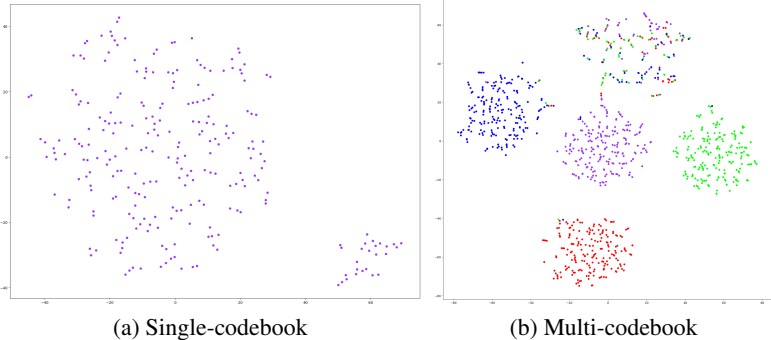

(a) Single-codebook           (b) Multi-codebook

Figure 3: Distribution Comparison. We employ t-SNE (Maaten & Hinton, 2008) to visualize the codewords in the codebook. It can be observed that the single-codebook method (a) compresses features from multiple source domains, resulting in the loss of domain-specific information. In contrast, the multi-codebook representation learning method (b) allocates a dedicated feature space for each source domain, mitigating the loss of domain-specific information caused by compression. (In the multi-codebook visualization, red, green, and blue represent the codebook distributions for the P, IMG, and Canon data branches, respectively, while purple indicates the distribution of the domain-invariant codebook.)

$$p' = (\frac{p - \mu^t}{\sigma^t})\sigma^s + \mu^s \tag{1}$$

where $\mu^s$, $\mu^t$, $\sigma^s$ and $\sigma^t$ represent the mean and variance of the source domain and target domain, respectively, while $p$ and $p'$ denote the target domain sample distributions before and after transfer.

**Solution 1:** We introduce a codebook into the representation learning of source domain features. During inference, pixel-level optimal codeword matching and replacement are performed on the target domain features, thereby achieving highly fine-grained feature transfer of target domain samples.

**Challenge 2: Loss of domain-specific feature information**

Existing domain generalization methods employ a single backbone network or codebook to learn representations from multiple diverse source domain distributions. However, as shown in Figure 3, such methods compress these diverse source distributions into a compact feature space, leading to the loss of domain-specific features.

**Solution 2:** We extend the single codebook to a multi-codebook framework, comprising a shared codebook (referred to as the domain-invariant codebook) for learning domain-invariant features, and multiple domain-specific codebooks dedicated to capturing domain-specific features. This method not only effectively preserves detailed information from each source domain, but also enhances the robustness of deep networks through a disentangled learning strategy that separates domain-specific and domain-invariant features. (Solution 2 extends Solution 1 and is referred to as Representation Learning Strategy based on Multiple Codebooks, RLMC.)

**Challenge 3: Optimal selection of domain-specific codebooks**

The introduction of multiple domain-specific codebooks necessitates the selection of the most appropriate domain-specific codebook for the target domain features during testing. Existing methods in related scenarios, such as Mixture of Experts (MoE), typically employ a gating mechanism—often implemented as a classification network. However, these methods inadequately account for domain shift. As a result, the classification network trained on source domain samples performs poorly when applied to target domain samples with different distributions (as shown in Table 3), leading to suboptimal codebook selection and misguided representation of domain-specific features in the target domain.

**Solution 3:** We propose a voting-based strategy for optimal domain-specific codebook selection, guided by multiple source domain-specific features. This method effectively reduces the inaccuracy of the classification network through multi-party voting.

Our main contributions can be summarized as follows:

- We introduce a codebook to perform pixel-level, highly fine-grained feature transfer on the target domain samples, significantly enhancing the efficiency of feature transfer for low-level vision tasks (SR tasks). To the best of our knowledge, the proposed method is the first test-time domain generalization method designed for low-level vision tasks, and also the first to incorporate a codebook into test-time domain generalization.
- We propose a Representation Learning Strategy based on Multiple Codebooks (RLMC), which mitigates the loss of domain-specific information from the source domains. Additionally, to address the challenge of selecting the optimal domain-specific codebook during testing, we introduce a voting-based strategy for optimal domain-specific codebook selection.
- Extensive experiments have been conducted to evaluate the proposed method across various network architectures and diverse sample distributions. The results demonstrate that our method significantly improves the efficiency of feature transfer in target domains for low-level vision tasks.

Refer to the Appendix A for related work on codebook and TTDG methods.

## 2 PROPOSED METHOD

### 2.1 PROBLEM SETTING

Given a source domain set $D^s$ consisting of multiple source domains with diverse data distributions, denoted as $D^s = \{D_1^s, D_2^s, \ldots, D_{n_s}^s\}$. Each source domain comprises paired LR and HR images: $D_i^s = \{LR_j^s, HR_j^s\}_{j=1}^{n_{ss}}$. Similarly, the target domain set $D^T$ contains multiple target domains with varying distributions: $D^T = \{D_1^T, D_2^T, \ldots, D_{n_t}^T\}$. Unlike the source domains, each target domain contains only LR images: $D_i^T = \{LR_j^T\}_{j=1}^{n_{tt}}$, and the source and target domain sets are disjoint: $D^s \cap D^T = \emptyset$. $n_s$ and $n_T$ represent the number of source domains and target domains, respectively. $n_{ss}$ and $n_{tt}$ denote the number of samples in the source domain and target domain, respectively. A model $M^s$ is trained on the source domain set $D^s$.

Consistent with existing TTDG methods, our goal is to transfer the distribution of the target samples ($LR^T$) to approximate that of the source domain samples ($LR^s$), thereby enhancing the performance of the source domain-trained model $M^s$ on the target domain.

### 2.2 OVERVIEW

As shown in Figure 4, the overall pipeline of the proposed method consists of two stages: During the training stage on the server side, pre-trained network weights (comprising Conv1, Backbone, and Decoder, which together form a complete network architecture) are first loaded and frozen. The proposed Representation learning strategy based on multiple codebooks (RLMC) (Section 2.3) is then applied to implicitly learn both domain-invariant features and domain-specific features from the source domains. (It is worth noting that the primary optimization objective during the training stage is to endow RLMC with the ability to reconstruct source domain features.) In the testing stage on the edge device side (Section 2.4), the learned domain-invariant codebook is utilized to transfer the domain-invariant features within the target domain sample distribution. Meanwhile, a voting-based optimal domain-specific codebook selection strategy (Section 2.5) is employed to choose the most suitable domain-specific codebook for the target domain samples, followed by corresponding transfer of the domain-specific features in the target domain.

### 2.3 TRAINING STAGE (REPRESENTATION LEARNING STRATEGY BASED ON MULTIPLE CODEBOOKS, RLMC)

Existing methods typically employ a single backbone network or a single codebook to represent features from multiple source domains with different distributions. This practice compresses the features of various domains into a constrained space (as shown in Figure 3), resulting in the loss of

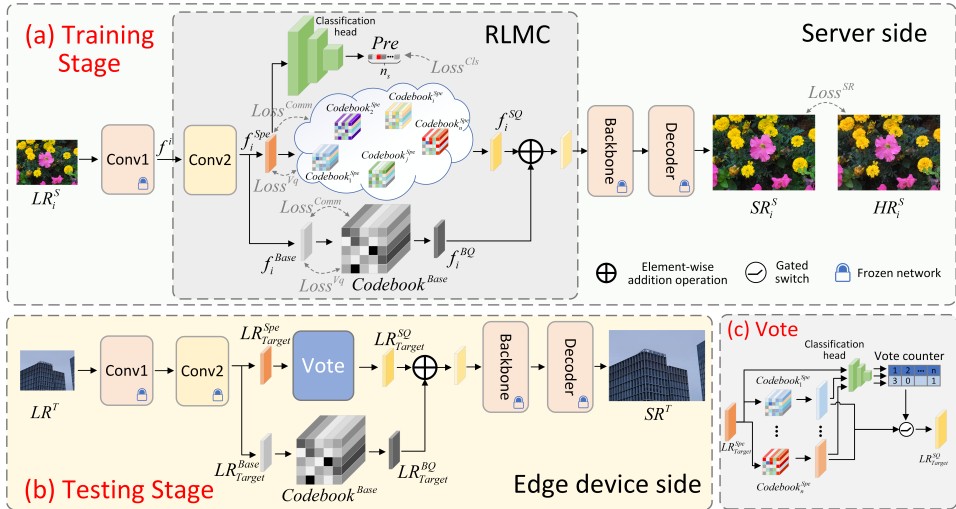

Figure 4: Overall Framework. $n_s$ denotes the number of source domains. $LR_i^s$, $SR_i^s$ and $HR_i^s$ denote the LR image, super-resolution image, and ground-truth HR image of the $i$-th source domain, respectively. $LR^T$ and $SR^T$ represent the LR image and super-resolution image of the target domain. $Codebook^{Base}$ and $Codebook^{Spe}$ refer to the domain-invariant codebook and domain-specific codebook, respectively. Definitions of other variables can be found in Equations 1 to 17.

domain-specific features. To mitigate this issue, we introduce multiple domain-specific codebooks to represent and learn features from the diverse source domains, thereby avoiding feature loss caused by excessive compression of source domain features.

Additionally, to enhance the network's efficiency in learning both domain-invariant and domain-specific features, we introduce an additional shared codebook, termed the domain-invariant codebook. Using this domain-invariant codebook as a base representation and the domain-specific codebooks as offsets, each source domain can be represented efficiently and discriminatively.

As shown in Figure 4(a), the LR image subset of the i-th source domain, denoted as $LR_i^S$, is first fed into a shallow feature extractor (Conv 1) to obtain shallow features $f_i$. These features are then decomposed via convolutional operations (Conv 2) into $f_i^{Base}$ and $f_i^{Spe}$. The feature $f_i^{Base}$ is transferred using the shared codebook (Domain-invariant codebook), while $f_i^{Spe}$ is transferred via the domain-specific codebook. This method implicitly guides the convolutional (Conv 2) decomposition such that $f_i^{Base}$ captures domain-invariant features—encoding cross-domain shared features—and $f_i^{Spe}$ retains domain-specific features.

$$f_i = Conv1(LR_i^S), \tag{2}$$

$$f_i^{Base} = Conv2(f_i), \tag{3}$$

$$f_i^{Spe} = f_i - f_i^{Base}, \tag{4}$$

$$f_i^{BQ} = Transfer(f_i^{Base}, Codebook^{Base}), \tag{5}$$

$$f_i^{SQ} = Transfer(f_i^{Spe}, Codebook_i^{Spe}), \tag{6}$$

where both $Conv1$ and $Conv2$ denote convolutional operations, and $Transfer$ refers to the feature transfer operation. Further details of the transfer process are elaborated in Equation 15. $Codebook^{Base}$ denotes the domain-invariant codebook, while $Codebook_i^{Spe}$ represents the domain-specific codebook corresponding to the $i$-th source domain. Both codebooks consist of multiple codewords, with $Codebook^{Base} = \{Codeword_j \in \mathbb{R}^D\}_{j=1}^{N_c}$, where $N_c$ is the number of codewords in $Codebook^{Base}$, and $D$ denotes the dimensionality of each codeword, which matches the number of channels in the target domain sample features. The structure of $Codebook_i^{Spe}$ is defined similarly to that of $Codebook^{Base}$.

The transferred domain-invariant features $f_i^{BQ}$ and domain-specific features $f_i^{SQ}$ are combined via element-wise summation and then passed into subsequent network layers to generate the corresponding SR image $SR_i^S$,

$$SR_i^S = Decoder(Backbone(f_i^{BQ} + f_i^{SQ})), \tag{7}$$

where $Backbone$ and $Decoder$ denote the backbone network and the decoder module, respectively.

Additionally, we introduce an auxiliary classification branch that utilizes the input domain-specific features $f_i^{Spe}$ to identify the category of the current domain (as formulated in Equation 8). It is worth noting that the predictions from this classification network are not utilized during the training stage.

$$Pre = CN(f_i^{Spe}), \tag{8}$$

where $Pre \in \mathbb{R}^{B,n_s}$ represents the category information predicted by the classification network $CN$, with $B$ and $n_s$ denoting the batch size and the number of source domains, respectively.

The overall training loss consists of four components: the image SR loss $Loss^{SR}$, the quantization loss $Loss^{vq}$, the commitment loss $Loss^{comm}$, and the classification loss $Loss^{Cls}$, formulated as follows:

$$Loss^{All} = Loss^{SR} + \lambda Loss^{Vq} + \beta Loss^{Comm} + \gamma Loss^{Cls}, \tag{9}$$

$$Loss^{SR} = |SR_i^S - HR_i^S|, \tag{10}$$

$$Loss^{Vq} = ||sg(f_i^{Base}) - f_i^{BQ}|| + ||sg(f_i^{Spe}) - f_i^{SQ}||, \tag{11}$$

$$Loss^{Comm} = ||f_i^{Base} - sg(f_i^{BQ})|| + ||f_i^{Spe} - sg(f_i^{SQ})||, \tag{12}$$

$$loss^{Cls} = CrossEntropy(Pre, i), \tag{13}$$

where $SR_i^S$ and $HR_i^S$ denote the predicted SR image and the corresponding ground-truth HR image of the $i$-th source domain, respectively. The operator $sg()$ represents the gradient stop operation. $Loss^{vq}$ denotes the quantization loss, which is used to train the codewords in the codebook, while $Loss^{comm}$ constrains the network layers to align with the codewords. The terms $\lambda$, $\beta$, and $\gamma$ are constant weighting coefficients.

## 2.4 TESTING STAGE

Existing test-time domain generalization methods typically rely on global style transfer to align the feature distribution of the target domain with that of the source domain. However, such global adjustment operates at a low granularity, making it difficult to perform effective feature transfer for pixel-level tasks like image SR. To address this limitation, we employ a codebook to represent the source domain and propose a nearest-neighbor matching strategy to achieve pixel-level transfer of target domain samples.

As shown in Figure 4(b), the target domain image $LR^T$ is fed into a shallow feature extractor (Conv1) to generate shallow features. Convolutional operations (Conv2) then decompose these shallow features into domain-invariant features and domain-variant features. The domain-invariant codebook is utilized to perform pixel-level feature transfer on the domain-invariant features of the target domain samples,

$$f_{Target}^{BQ} = Transfer(f_{Target}^{Base}, Codebook^{Base}), \tag{14}$$

where $f_{Target}^{Base} \in \mathbb{R}^{B,C,H,W}$, with $B$, $C$, $H$, and $W$ denoting the batch size, number of channels, height, and width of the feature, respectively.

The feature transfer operation computes the most similar (nearest-neighbor) codeword in the codebook to each pixel feature and replaces the corresponding target domain feature at that pixel location using the retrieved codeword. Taking the transfer of the pixel feature $f_{x,y}$ at position $x, y$ as an example, where $x \in \{1, 2, \ldots, H\}$ and $y \in \{1, 2, \ldots, W\}$, the procedure is as follows:

$$Transfer(f_{x,y}, Codebook) = C_k, \quad where \quad k = \underset{C_i \in Codebook}{argmini} ||f_{x,y} - C_i||_2, \tag{15}$$

where $C_k$ and $C_i$ represent the $k$-th and $i$-th codewords in the codebook.

For the domain-specific features, we employ the proposed voting-based domain-specific codebook selection strategy (Section 2.5) to choose the optimal domain-specific codebook, which is then used to perform feature transfer on the domain-specific features,

$$f_{Target}^{SQ} = Vote(f_{Target}^{Spe}), \tag{16}$$

where $Vote$ denotes the proposed voting-based domain-specific codebook selection strategy.

Finally, the transferred domain-specific features and domain-invariant features are combined via element-wise summation and fed into subsequent network layers to generate the SR image $SR^T$ for the target domain.

$$SR^T = Decoder(Backbone(f_{Target}^{BQ} + f_{Target}^{SQ})). \tag{17}$$

## 2.5 VOTING-BASED DOMAIN-SPECIFIC CODEBOOK SELECTION STRATEGY

The presence of multiple domain-specific codebooks raises the challenge of selecting the optimal codebook for feature transfer of target domain-specific features. Existing methods in related scenarios, such as Mixture of Experts (MoE) (Yang et al., 2025; Zhu et al., 2025; Zamfir et al., 2024), fail to account for domain shift. These methods typically employ a classification network to discriminate among experts and select the one with the highest confidence score as the optimal choice. However, we observe that due to the distributional discrepancy between the target and source domains, the classification network trained on source domain samples tends to yield significant prediction errors (Table 3). This leads to the selection of a suboptimal domain-specific codebook, consequently guiding the transfer of target domain-specific features in an erroneous direction.

Therefore, we propose to leverage multiple domain-specific codebooks to perform preliminary transfer on the target domain features, generating multiple sample features that approximate the distribution of the source domain features. Since the quality of these transferred features varies, we introduce a voting strategy to stabilize the predictions of the classification network.

As shown in Figure 4(c), the target domain-specific features are transferred using multiple source domain-specific codebooks and then fed into the classification network. The highest domain category corresponding to each transferred feature is counted. The domain-specific codebook with the highest number of votes is selected as the final choice. In the event of a tie (i.e., equal votes for multiple domain categories), the voting result of the original untransferred feature is adopted as the final prediction. The overall procedure is summarized in Appendix B.

## 3 EXPERIMENTS

### 3.1 EXPERIMENT DETAILS

**Datasets, Evaluation, and Network Architecture:** We extensively validate the proposed method on the DRealSR (Wei et al., 2020), Set5 (Bevilacqua et al., 2012), Set14 (Zeyde et al., 2010), B100 (Martin et al., 2001), Urban (Huang et al., 2015), Manga109 (Matsui et al., 2017), and DIV2K (Timofte et al., 2017) datasets. The DRealSR (Wei et al., 2020) dataset comprises multiple data branches—P, IMG, Canon, Pan, Sony, and DSC—each captured by different cameras with distinct data distributions, making it widely adopted in domain generalization and domain adaptation research. For performance evaluation, we employ PSNR, SSIM, and LPIPS metrics. To demonstrate the architectural generalization capability of the proposed method, we further verify its effectiveness on the AdaCode (Liu et al., 2023), HAT (Chen et al., 2023), and MambaIR (Guo et al., 2024a) network architectures.

**Training Details:** We train the network using the Adam optimizer with an input image size of $48 \times 48$ pixels. The batch size is set to 16. The training is conducted on $4 \times$ V100 GPUs.

### 3.2 ABLATION EXPERIMENT

### 3.2.1 ABLATION EXPERIMENTS ON DIFFERENT CODEBOOK SETTINGS.

To validate the effectiveness of the proposed method, we conducted an ablation study on the codebook configuration. As shown in Table 1, we used the performance of the MambaIR (Guo et al.,

Table 1: Ablation experiments on different codebook settings

| Method | Pan | | Sony | | DSC | |
|---|---|---|---|---|---|---|
| | PSNR↑ | SSIM↑ | PSNR↑ | SSIM↑ | PSNR↑ | SSIM↑ |
| Baseline | 31.0263 | 0.8580 | 30.7220 | 0.8645 | 30.9117 | 0.8816 |
| One codebook | 31.1111 | 0.8573 | 31.1411 | 0.8750 | 30.9640 | 0.8776 |
| w/o Domain invariant codebook | 30.9097 | 0.8552 | 31.0144 | 0.8749 | 30.9043 | 0.8782 |
| Domain specific codebook & Domain invariant codebook | 31.1571 | 0.8594 | 31.2937 | 0.8791 | 31.2118 | 0.8836 |

Table 2: Ablation experiments on different feature transfer methods

| Methods | Pan | | Sony | | DSC | |
|---|---|---|---|---|---|---|
| | PSNR↑ | SSIM↑ | PSNR↑ | SSIM↑ | PSNR↑ | SSIM↑ |
| Baseline | 31.0263 | 0.8580 | 30.7220 | 0.8645 | 30.9117 | 0.8816 |
| One center style transfer | 31.0262 | 0.8580 | 30.7171 | 0.8644 | 30.9131 | 0.8816 |
| Multi center style transfer | 31.0261 | 0.8580 | 30.7166 | 0.8644 | 30.9118 | 0.8816 |
| Codebook-based transfer (Ours) | 31.1571 | 0.8594 | 31.2937 | 0.8791 | 31.2118 | 0.8836 |

2024a) network—trained on the source domain set (P, IMG, and Canon data branches)—on the test domain as the baseline.

We first evaluated the method that employs only a single codebook to represent all source domain features (One Codebook). It can be observed that the codebook-based representation learning method effectively improves the network's performance on the target domain. By performing pixel-level feature transfer on the target domain samples, the distribution of the target domain can be transferred with relative effectiveness. However, using a single codebook to represent multiple source domains inevitably leads to the loss of domain-specific information (as shown in Figure 3), while the performance metrics are also lower compared to the multi-codebook method.

To validate the effectiveness of the domain-invariant codebook, we evaluated the performance of the SR network in the absence of this component (denoted as w/o Domain-Invariant Codebook), where only a domain-specific codebook was used for each source domain. However, such methods overlook the inherent correlations among source domains and weaken the network's ability to discriminate between domain-specific and domain-invariant features.

The proposed feature representation learning method, which jointly leverages domain-invariant and domain-specific codebooks, enables fine-grained feature representation for each source domain and avoids the loss of domain-specific information associated with single-codebook designs. By disentangling the representations of domain-invariant and domain-specific features, the method enhances the robustness of the domain-invariant codebook to diverse input features, while the domain-specific codebook becomes more specialized in capturing domain-specific features, leading to improved efficiency in target domain feature transfer. The proposed method achieves notable performance gains across all three data branches, with PSNR improvements of 0.1308 dB, 0.5717 dB, and 0.3001 dB on the Pan, Sony, and DSC branches, respectively.

### 3.3 ABLATION EXPERIMENTS ON DIFFERENT FEATURE TRANSFER METHODS

As shown in Table 2, we evaluated the transfer efficiency of different feature transfer methods on the target domain features. First, we tested the single-center style transfer method, which employs a unified style center across all source domains and adjusts the target domain distribution via mean and variance matching (as formulated in Equation 1). We also evaluated the multi-center style transfer method, where each source domain is represented by its domain-specific mean and variance. During inference, the nearest source domain style is selected to transfer the target domain features. The results indicate that style transfer is highly inefficient for low-level vision tasks, with almost no improvement in performance metrics.

In high-level vision tasks, the features learned by the network represent abstract interpretations of the entire image, and predictions rely heavily on the global style of these features. Style transfer can thus effectively influence the model's predictions. However, low-level vision tasks require pixel-wise

Table 3: Ablation experiments on different domain-specific codebook selection methods

|  | Pan | | Sony | | DSC | |
| --- | --- | --- | --- | --- | --- | --- |
| Methods | Top1↑ | Top2↑ | Top1↑ | Top2↑ | Top1↑ | Top2↑ |
| Maximum predicted score | 0.2907 | 0.5673 | 0.0596 | 0.1391 | 0.1660 | 0.3868 |
| Voting selection (Ours) | 0.3726 | 0.7838 | 0.3545 | 0.8757 | 0.4922 | 0.7800 |

Table 4: Performance comparison with other methods

|  | Pan | | | Sony | | | DSC | | |
| --- | --- | --- | --- | --- | --- | --- | --- | --- | --- |
| Methods | PSNR↑ | SSIM↑ | LPIPS↓ | PSNR↑ | SSIM↑ | LPIPS↓ | PSNR↑ | SSIM↑ | LPIPS↓ |
| TF-Cal (Zhao et al., 2022) | 28.85 | 0.7862 | 0.4318 | 27.92 | 0.7787 | 0.4387 | 29.51 | 0.8485 | 0.3690 |
| TSB (Park et al., 2023) | 30.15 | 0.8315 | 0.4155 | 29.28 | 0.8044 | 0.4309 | 30.78 | 0.8739 | 0.3709 |
| DG-PIC (Jiang et al., 2024) | 29.71 | 0.8135 | 0.4130 | 30.06 | 0.8305 | 0.4455 | 29.88 | 0.8470 | 0.4463 |
| TTDG (Zhou et al., 2024) | 29.71 | 0.8135 | 0.4463 | 30.06 | 0.8305 | 0.4455 | 29.88 | 0.8470 | 0.4130 |
| TTMG (Nam & Lee, 2025) | 30.26 | 0.8411 | 0.4523 | 30.56 | 0.8541 | 0.4069 | 30.70 | 0.8776 | 0.4128 |
| MC-TTDG (Ours) | **31.15** | **0.8594** | **0.3593** | **31.29** | **0.8791** | **0.3157** | **31.21** | **0.8836** | **0.3583** |

predictions, depending on fine-grained features of each individual pixel. Style transfer, operating at a coarse granularity, fails to transfer pixel-level features effectively. In contrast, the proposed codebook-based feature transfer strategy computes the nearest codeword in the codebook for each pixel feature and replaces the target domain feature at the corresponding pixel location using the retrieved codeword, thereby achieving extremely high granularity in feature transfer. As a result, the codebook-based method proves far more efficient for low-level vision tasks.

### 3.4 ABLATION EXPERIMENTS ON DIFFERENT DOMAIN-SPECIFIC CODEBOOK SELECTION METHODS

As shown in Table 3, we evaluated the accuracy of different strategies for selecting the optimal domain-specific codebook. We first tested the performance of each target domain image when using different domain-specific codebooks, then ranked the results. The domain-specific codebook that yielded the highest performance was defined as the optimal one.

Subsequently, we evaluated the accuracy of directly using a classification network to select the optimal domain-specific codebook—that is, choosing the codebook with the highest prediction score. However, due to the distribution shift between the target and source domains, the classification network trained on the source domain exhibits low accuracy on the target distribution. This leads to misguided transfer of the target features and poor performance of the neural network on the target domain (Line 2 of Appendix Table 5).

The proposed voting-based strategy for optimal domain-specific codebook selection leverages multiple domain-specific codebooks to perform preliminary transfer on the target domain features, thereby facilitating more reliable predictions by the classification network. By aggregating multiple predictions through voting, this method enhances the accuracy of the selection process.

### 3.5 COMPARATIVE EXPERIMENT

As shown in Table 4, we implemented existing test-time domain generalization methods for low-level vision tasks, including TF-Cal (Zhao et al., 2022), TSB(Park et al., 2023), DG-PIC(Jiang et al., 2024), TTDG(Zhou et al., 2024), and TTMG(Nam & Lee, 2025). These methods employ style transfer—an operation with coarse granularity—for target domain feature distribution transfer. Consequently, they are unsuitable for low-level vision tasks requiring pixel-level predictions. In contrast, the proposed MC-TTDG method utilizes a shared domain-invariant codebook and domain-specific codebooks to achieve pixel-level feature representation and transfer, significantly enhancing the performance of SR networks on the target domain.

## 4 CONCLUSION

In this paper, we propose a test-time domain generalization method for SR tasks, which employs a codebook strategy to achieve pixel-level transfer of target domain sample features. Our proposed representation learning strategy based on multiple codebooks utilizes a domain-invariant codebook

and multiple domain-specific codebooks to enable fine-grained representation of source domain samples. This method not only prevents the loss of domain-specific features but also enhances the robustness of the domain-invariant codebook and improves the domain specificity of the domain-specific codebooks. Furthermore, the introduced voting-based strategy for optimal domain-specific codebook selection effectively mitigates the inaccuracy of the classification network caused by domain shift through multi-party voting, thereby providing stable and appropriate transfer directions for target domain sample features.

## ACKNOWLEDGMENTS

This work was supported by the Fundamental and Interdisciplinary Disciplines Breakthrough Plan of the Ministry of Education of China (Grant JYB2025XDXM118), and the Natural Science Foundation of China (Grant 62176119).

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

# A  RELATED WORK

## A.1  CODEBOOK

Chen et al. (2022) first proposed a codebook-based SR network. During training, high-resolution (HR) image features were quantized and stored in the codebook. At inference, the network retrieved quantized HR features that closely matched the input low-resolution (LR) image features, significantly improving the SR performance of the SR network. Zhou et al. (2022) proposed a codebook-based face reconstruction network that utilized a Transformer to predict the optimal codebook entry for each pixel feature. They further introduced a residual structure to incorporate LR image features into the decoding module, effectively preventing the loss of texture details. Liu et al. (2023) argued that existing methods employed a single codebook for all scenarios, where one codebook needed to simultaneously encode diverse scenes (e.g., indoor, outdoor, and facial images), resulting in feature confusion. To address this, they proposed a multi-codebook strategy that assigned a dedicated codebook to each specific scene, significantly improving the feature encoding efficiency of the codebooks. Lu et al. (2024) addressed the image compression-decompression problem by proposing a HybridFlow dual-stream framework. This method combined a discrete stream based

on high-quality codebook (ensuring perceptual quality) with an ultra-low bitrate continuous feature stream, achieving high-fidelity image reconstruction at high compression ratios. Zhang et al. (2024) incorporated codebook into vision-language models, where the codebook effectively modeled the correlation between image features and semantic features, thereby significantly enhancing the robustness of VLM models. Li et al. (2024b) introduced codebook into the fake face detection task by comparing the discrepancies between codebook-reconstructed faces and authentic faces, which enabled effective identification of synthetic faces. Wen et al. (2025) identified that existing methods faced two key limitations: (1) nearest-neighbor-based feature matching methods suffered from erroneous feature matching, and (2) multi-codebook methods tended to produce over-smoothed features. To address these issues, they first employed Top-K feature matching to retrieve the K most relevant codebooks, then performed feature fusion across these K codebooks before conducting final feature matching. Wu et al. (2025) introduced codebook into the low-light image enhancement task. Long et al. (2025) applied codebook to domain generalization tasks, where they theoretically demonstrated that discretized features from codebook exhibited superior robustness compared to continuous features.

Although a few multi-codebook methods (Liu et al., 2023; Wen et al., 2025) have been proposed, they learned representations **sequentially**—training only one codebook at a time for each source domain. Furthermore, these methods did not explicitly incorporate the concepts of domain-specific and domain-invariant features. In contrast, the proposed method (MC-TTDG) performed representation learning **simultaneously** across multiple source domains (i.e., training all codebooks concurrently). It implicitly disentangled the features into domain-specific and domain-invariant representations, which not only enhanced the robustness of the network but also significantly reduces the training time required for the codebooks.

## A.2 DOMAIN GENERALIZATION

Domain generalization methods enhanced model robustness by either explicitly or implicitly learning domain-invariant and domain-specific features (Hu et al., 2023; Guo et al., 2023; Vidit et al., 2023; Li et al., 2024a; Cheng et al., 2024), or by constraining the consistency between network outputs before and after augmentation (Chattopadhyay et al., 2023; Wang et al., 2024; Long et al., 2024; Guo et al., 2024b; Ahn et al., 2024; Danish et al., 2024), thereby improving the network's generalization capability against input degradations.

The Test-Time Domain Generalization (TTDG) method enhanced the performance of a source domain-trained network on target domains by aligning the target domain distribution with the source distribution during testing, while circumventing the substantial computational overhead associated with fine-tuning on target domain samples.

Zhao et al. (2022) postulated that amplitude features extracted through Fourier analysis represent domain-specific features. Consequently, they transferred the amplitude features of target domain samples toward the centroid of the source domain's amplitude distribution. Jiang et al. (2022) employed a meta-learning framework to partition the source domain into a meta-source domain and a meta-target domain. Using individual meta-target domain samples along with the meta-source domain, they predicted the overall distribution (mean and variance) of the meta-target domain. This method enabled the estimation of the real target domain's distribution, thereby facilitating feature normalization. Park et al. (2023) addressed the issue of imbalanced sample sizes across source domains by first performing category-aware transfer between these domains. During testing, style transfer was applied exclusively to target domain samples that exhibited a large distributional distance from the source domains, while those closer to the source domains were left unmodified. Jiang et al. (2024) incorporated both global and local style transfer by applying global pooling to source domain point clouds to extract global features, while also dividing the point clouds into patches to capture local features. Through dual global and local adjustment of target domain point clouds, this approach effectively enhanced the efficiency of style transfer for target domain data. Yu & Hwang (2024) trained expert prompts using contextual image data and enhanced test samples by incorporating these pre-trained prompts to guide the network toward accurate image classification. Zhou et al. (2024) imposed constraints on the styles generated from the source domain to ensure orthogonality among them, thereby promoting diversity. Ma et al. (2024) reduced the distributional distance between the target and source domains by applying noisy masks to target domain samples. Meng et al. (2025) projected both source domain and target domain into a shared feature space, which

enhanced the performance of deep neural networks on the target domain. Nam & Lee (2025) identified domain-specific features through covariance analysis and adjusted the feature distribution of the target domain via style transfer.

The aforementioned methods primarily targeted high-level vision tasks such as image and point cloud classification, where features represent abstract interpretations of the entire image. By applying style transfer—namely, adjusting the mean and variance—these methods successfully transferred key predictive features in the target domain to align with the source domain distribution. However, such global style transfer proved ineffective for low-level vision tasks like image SR, which require per-pixel prediction and are sensitive to fine-grained local features. To address this limitation, we introduced a codebook-based framework into test-time domain generalization. Specifically, we constructed a domain-specific codebook to achieve high-fidelity representation of each source domain at a fine-grained level. During testing, pixel-level feature transfer was applied to the target domain samples, which significantly improved both the efficiency of distribution transfer and the performance of deep networks on the target domain for low-level vision tasks.

## B   ALGORITHM

As shown in Algorithm 1, we provide a detailed presentation of the voting-based domain-specific codebook selection strategy using pseudocode.

---

**Input:** Classification network $CN$; domain-specific codebooks $Codebook^{Spe} \in \{Codebook_i^{Spe}\}_{i=1}^{n_s}$; domain-specific features $f_{Target}^{Spe}$; Vote counter $Co \in \mathbb{R}^{n_s}$, where $Co$ is initialized as an all-zero vector;
**Output:** The transferred domain-specific features $f_{Target}^{SQ}$;

1 **for** $i \leftarrow 1$ **to** $n_s$ **do**
2     $f_i^{SQ} = Transfer(f_{Target}^{Spe}, Codebook_i^{Spe})$;
3     $Pre_i = CN(f_i^{SQ})$;
4     $Index_i = Argmax(Pre_i)$;
5     $Co[Index_i] \mathrel{+}= 1$;
6 **end**
7 **if** *A tie in votes* **then**
8     $Pre = CN(f_{Target}^{Spe})$
9     $Index_{Best} = Argmax(Pre)$
10 **end**
11 **else**
12     $Index_{Best} = Argmax(Co)$
13 **end**
14 $f_{Target}^{SQ} = Transfer(f_{Target}^{Spe}, Codebook_{Index_{Best}}^{Spe})$
15 **return** $f_{Target}^{SQ}$;

**Algorithm 1:** Voting-based domain-specific codebook selection strategy

---

## C   ADDITIONAL ABLATION EXPERIMENTS

### C.1   ABLATION EXPERIMENTS ON DIFFERENT DOMAIN-SPECIFIC CODEBOOK SELECTION METHODS

Table 5: Ablation experiments on different domain-specific codebook selection methods

| Methods | Pan | | Sony | | DSC | |
|---|---|---|---|---|---|---|
| | PSNR↑ | SSIM↑ | PSNR↑ | SSIM↑ | PSNR↑ | SSIM↑ |
| Baseline | 31.0263 | 0.8580 | 30.7220 | 0.8645 | 30.9117 | 0.8816 |
| Maximum predicted score | 31.0495 | 0.8584 | 30.6124 | 0.8667 | 30.9339 | 0.8799 |
| Minimum distance | 31.0203 | 0.8573 | 30.6992 | 0.8675 | 30.9578 | 0.8794 |
| Voting selection (Ours) | 31.1571 | 0.8594 | 31.2937 | 0.8791 | 31.2118 | 0.8836 |

Table 6: Ablation experiments on domain-specific features

| Methods | Pan | | Sony | | DSC | |
|---|---|---|---|---|---|---|
| | PSNR↑ | SSIM↑ | PSNR↑ | SSIM↑ | PSNR↑ | SSIM↑ |
| Baseline | 31.0263 | 0.8580 | 30.7220 | 0.8645 | 30.9117 | 0.8816 |
| w/o domain-specific feature | 30.3096 | 0.8452 | 30.8673 | 0.8743 | 30.2964 | 0.8671 |
| w/o domain-specific feature transfer | 30.9597 | 0.8586 | 30.9966 | 0.8726 | 31.1124 | 0.8831 |
| Ours | 31.1571 | 0.8594 | 31.2937 | 0.8791 | 31.2118 | 0.8836 |

As shown in Table 5, we evaluate the performance of the SR network under different domain-specific codebook selection strategies. First, we test the performance when using a classification network to select the optimal domain-specific codebook (denoted as Maximum Predicted Score). Due to the significant domain shift between the target domain and source domain distributions, the classification network trained on the source domain exhibits low accuracy on target domain samples (Line 1 of Table 3). This results in incorrect transfer of the target sample distribution, ultimately degrading the performance of the SR network on the target domain.

Additionally, we evaluated the method of directly computing the minimum-distance codeword across all codebooks for each target domain pixel feature and replacing the current pixel feature accordingly (Minimum distance). However, this method demonstrated poor performance, as the fusion of features from multiple domains disrupts the prediction behavior of the network.

Our proposed voting-based domain-specific codebook selection strategy first pre-transfers the target domain distribution to approximate that of the source domain. Through multi-party voting, it significantly enhances the accuracy of domain-specific codebook selection (Line 2 of Table 3), thereby effectively improving the performance of the SR network on the target domain.

## C.2 ABLATION EXPERIMENTS ON DOMAIN-SPECIFIC FEATURES

As shown in Table 6, to validate the effectiveness of domain-specific features, we first evaluated the performance metrics of the network in the absence of these features (w/o domain-specific feature). As illustrated in the first row of Figure 5, the SR image becomes blurred when domain-specific features are removed, whereas incorporating them significantly enhances high-frequency details. This indicates that, for image SR networks, high-frequency textual details constitute a major part of domain-specific features. Introducing domain-specific features effectively improves the network's capability to reconstruct fine textual details. Furthermore, we conducted an ablation study on whether feature transfer is necessary for domain-specific features, specifically testing the performance when using untransferred domain-specific features directly (w/o domain-specific feature transfer). Since the target domain-specific features were not transferred to approximate the source domain distribution, the subsequent source-domain-trained network layers were incompatible with these features, leading to degraded performance.

## C.3 ABLATION EXPERIMENTS ON DOMAIN-INVARIANT AND DOMAIN-SPECIFIC FEATURE SEPARATION STRATEGIES

As shown in Table 7, we evaluate the impact of different separation strategies for domain-invariant and domain-specific features on network performance. Existing domain generalization methods for SR typically define high-frequency information as domain-specific features and low-frequency information as domain-invariant features, employing explicit separation strategies such as Fourier transforms, wavelet transforms, or downsampling/upsampling operations. However, due to distribution shifts between test and training samples, these pre-defined separation strategies often demonstrate poor adaptability to domain shift, frequently leading to incorrect feature partitioning. In contrast, our implicit feature separation strategy—implemented through one-to-one domain-specific codebooks for each source domain alongside a shared domain-invariant Codebook—enables the network to adaptively separate domain-invariant and domain-specific features. This approach demonstrates stronger adaptability to domain shift compared to explicit separation methods.

Table 7: Ablation experiments on domain-invariant and domain-specific feature separation strategies

| Methods | Pan | | Sony | | DSC | |
|---|---|---|---|---|---|---|
| | PSNR↑ | SSIM↑ | PSNR↑ | SSIM↑ | PSNR↑ | SSIM↑ |
| Downsampling/upsampling | 30.5356 | 0.8468 | 30.7316 | 0.8762 | 30.5960 | 0.8699 |
| Wavelet transforms | 30.6811 | 0.8479 | 31.0207 | 0.8746 | 30.8287 | 0.8737 |
| Fourier transforms | 31.1054 | 0.8548 | 30.9231 | 0.8767 | 30.1373 | 0.8792 |
| Ours | 31.1571 | 0.8594 | 31.2937 | 0.8791 | 31.2118 | 0.8836 |

Table 8: Ablation experiments on the number of domain-specific codebooks

| Number | Pan | | Sony | | DSC | |
|---|---|---|---|---|---|---|
| | PSNR↑ | SSIM↑ | PSNR↑ | SSIM↑ | PSNR↑ | SSIM↑ |
| Baseline | 31.0263 | 0.8580 | 30.7220 | 0.8645 | 30.9117 | 0.8816 |
| 1 | 31.0815 | 0.8582 | 31.0125 | 0.8708 | 31.0912 | 0.8824 |
| 2 | 31.1012 | 0.8586 | 31.2891 | 0.8732 | 31.1593 | 0.8831 |
| 3 | 31.1571 | 0.8594 | 31.2937 | 0.8791 | 31.2118 | 0.8836 |
| 5 | 31.1783 | 0.8592 | 31.3129 | 0.8795 | 31.2482 | 0.8834 |

## C.4 ABLATION EXPERIMENTS ON THE NUMBER OF DOMAIN-SPECIFIC CODEBOOKS

As shown in Table 8, we evaluate the effect of varying the number of domain-specific codebooks on network performance. Since MC-TTDG requires a one-to-one correspondence between source domains and domain-specific codebooks for implicit separation of domain-specific and domain-invariant features, and considering that the DRealSR dataset contains a maximum of six camera branches, we configure up to five camera branches as source domains and one as the target domain. The MC-TTDG method exhibits strong robustness to the number of domain-specific codebooks, consistently improving performance on the target domain across different codebook quantities.

## C.5 ABLATION EXPERIMENTS ON THE NUMBER OF CODEWORDS

As shown in Table 9, we evaluate the impact of varying the number of codewords on network performance. Reducing the number of codewords inevitably diminishes the representational capacity of the codebook, thereby lowering the transformation accuracy of target domain features. In contrast, increasing the number of codewords enhances the diversity of preserved source domain features in the codebook, leading to improved network performance metrics.

Table 9: Ablation experiments on the number of codewords

| Number | Pan | | Sony | | DSC | |
|---|---|---|---|---|---|---|
| | PSNR↑ | SSIM↑ | PSNR↑ | SSIM↑ | PSNR↑ | SSIM↑ |
| 128 | 31.1054 | 0.8589 | 31.1052 | 0.8710 | 31.1723 | 0.8821 |
| 256 | 31.1571 | 0.8594 | 31.2937 | 0.8791 | 31.2118 | 0.8836 |
| 512 | 31.1612 | 0.8599 | 31.3001 | 0.8788 | 31.2192 | 0.8841 |

## C.6 ABLATION EXPERIMENTS ON FINE-TUNING EXTENT

As shown in Table 10, we compare the effects of fine-tuning the entire network versus fine-tuning only the codebook on overall performance. Since the codebook is not initialized from pre-trained weights, full-network fine-tuning would allow the uninitialized parameters to disrupt the pre-trained backbone. In contrast, our MC-TTDG method keeps all other network layers frozen, which effectively preserves the backbone's stability while improving the codebook's learning efficiency for source domain features.

Table 10: Ablation experiments on Fine-tuning Extent

| | Pan | | Sony | | DSC | |
|---|---|---|---|---|---|---|
| Fine-tuning Extent | PSNR↑ | SSIM↑ | PSNR↑ | SSIM↑ | PSNR↑ | SSIM↑ |
| Entire network | 30.8873 | 0.8559 | 30.6771 | 0.8695 | 30.7417 | 0.8777 |
| Codebook | 31.1571 | 0.8594 | 31.2937 | 0.8791 | 31.2118 | 0.8836 |

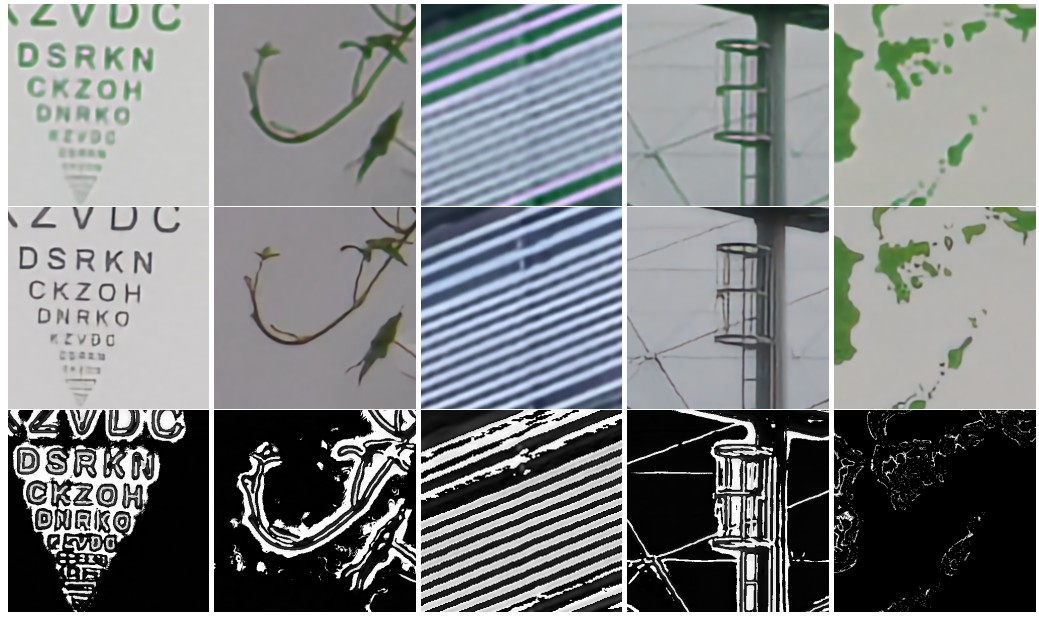

Figure 5: Visual comparison. The first and second rows show the images generated without and with the transferred domain-specific features, respectively. The third row displays a visual representation of the difference between the first and second rows.

## C.7 COMPUTATIONAL EFFICIENCY COMPARISON WITH TEST-TIME DOMAIN ADAPTATION METHODS

As shown in Table 11, we compare MC-TTDG with methods that similarly leverage test samples to enhance performance—such as test-time adaptation approaches (e.g., SRTTA, IODA, DASR) that fine-tune the network using test data. Our method demonstrates significantly lower computational demands. These alternative approaches require substantial training time, making network fine-tuning particularly challenging for edge devices in practical applications.

In contrast, MC-TTDG operates by aligning the target domain distribution with the source domain distribution. It performs only inference on target domain samples without network fine-tuning, substantially reducing computational overhead in real-world scenarios. This design makes MC-TTDG particularly suitable for resource-constrained environments compared to test-time domain adaptation methods.

Table 11: Computational Efficiency Comparison with Test-Time Domain Adaptation Methods. The computations were performed on a single V100 GPU with a patch size of 48×48. Abbreviations: FT (Fine-Tuning), TTDA (Test-Time Domain Adaptation), TTDG (Test-Time Domain Generalization).

| Methods | Methods | Time (Min) | Need Training |
|---|---|---|---|
| FT | Fine-Tuning | 1380 | Yes |
| TTDA | DASR (Wei et al., 2021) | 19.33 | Yes |
| | IODA (Tang & Yang, 2024) | 9.01 | Yes |
| | SRTTA (Deng et al., 2023) | 10.75 | Yes |
| TTDG | MC-TTDG | 0.00135 | No |

## C.8 ABLATION EXPERIMENTS ON DIFFERENT SAMPLE DISTRIBUTIONS

As shown in Table 12, we validated the effectiveness of the proposed method across diverse data distributions, including the Set5 (Bevilacqua et al., 2012), Set14 (Zeyde et al., 2010), B100 (Martin et al., 2001), Urban (Huang et al., 2015), Manga109 (Matsui et al., 2017), and DIV2K (Timofte et al., 2017) datasets. MC-TTDG achieves consistent performance improvements across all data distributions.

Table 12: Ablation experiments on different sample distributions

|  | B100 | | Mamga109 | | Set14 | |
| --- | --- | --- | --- | --- | --- | --- |
| Methods | PSNR↑ | SSIM↑ | PSNR↑ | SSIM↑ | PSNR↑ | SSIM↑ |
| Baseline | 23.5493 | 0.6216 | 22.1986 | 0.7440 | 23.3874 | 0.6426 |
| MC-TTDG | 23.8877 | 0.6260 | 22.6995 | 0.7482 | 23.7331 | 0.6507 |
|  | DIV2K | | Urban100 | | Set5 | |
| Methods | PSNR↑ | SSIM↑ | PSNR↑ | SSIM↑ | PSNR↑ | SSIM↑ |
| Baseline | 21.5490 | 0.5731 | 20.5615 | 0.5904 | 24.9639 | 0.7586 |
| MC-TTDG | 22.2340 | 0.6341 | 21.0803 | 0.6056 | 25.2720 | 0.7546 |

## C.9 ABLATION EXPERIMENTS ON DIFFERENT NETWORK ARCHITECTURES

The SR network consists of two branches: one prioritizing pixel fidelity and the other emphasizing visual perception. We conducted experimental analyses on both branches. As shown in Tables 13 and 14, the proposed method effectively enhances the performance of the SR network on the target domain.

Table 13: Ablation experiments on perception-oriented SR Networks

|  | Pan | | | Sony | | |
| --- | --- | --- | --- | --- | --- | --- |
| Methods | LPIPS↓ | Dists↓ | FID↓ | LPIPS↓ | Dists↓ | FID↓ |
| AdaCode (Liu et al., 2023) | 0.2688 | 0.1451 | 25.12 | 0.2755 | 0.1594 | 30.47 |
| AdaCode + MC | 0.2634 | 0.1438 | 23.75 | 0.2579 | 0.1450 | 31.39 |
|  | DSC | | | | | |
| Methods | LPIPS↓ | Dists↓ | FID↓ | | | |
| AdaCode (Liu et al., 2023) | 0.2480 | 0.1340 | 25.70 | | | |
| AdaCode + MC | 0.2417 | 0.1302 | 23.79 | | | |

Table 14: Ablation experiments on fidelity-Oriented SR Networks

|  | Pan | | Sony | | DSC | |
| --- | --- | --- | --- | --- | --- | --- |
| Methods | PSNR↑ | SSIM↑ | PSNR↑ | SSIM↑ | PSNR↑ | SSIM↑ |
| HAT(Chen et al., 2023) | 31.0249 | 0.8607 | 31.0376 | 0.8763 | 30.9813 | 0.8837 |
| HAT + MC | 31.0379 | 0.8593 | 31.1766 | 0.8779 | 31.1334 | 0.8826 |
| MambaIR (Guo et al., 2024a) | 31.0263 | 0.8580 | 30.7220 | 0.8645 | 30.9117 | 0.8816 |
| MambaIR + MC | 31.1571 | 0.8594 | 31.2937 | 0.8791 | 31.2118 | 0.8836 |

## D VISUALIZATION RESULTS

As shown in Figure 6 and Figure 7, we compared the proposed MC-TTDG with other test-time domain generalization methods. MC-TTDG demonstrated superior detail restoration and noise suppression.

## E STATEMENT ON LLM USAGE

We used a Large Language Model (LLM), specifically ChatGPT, solely for language polishing and improving the readability of the manuscript. The LLM was not used to generate ideas, conduct

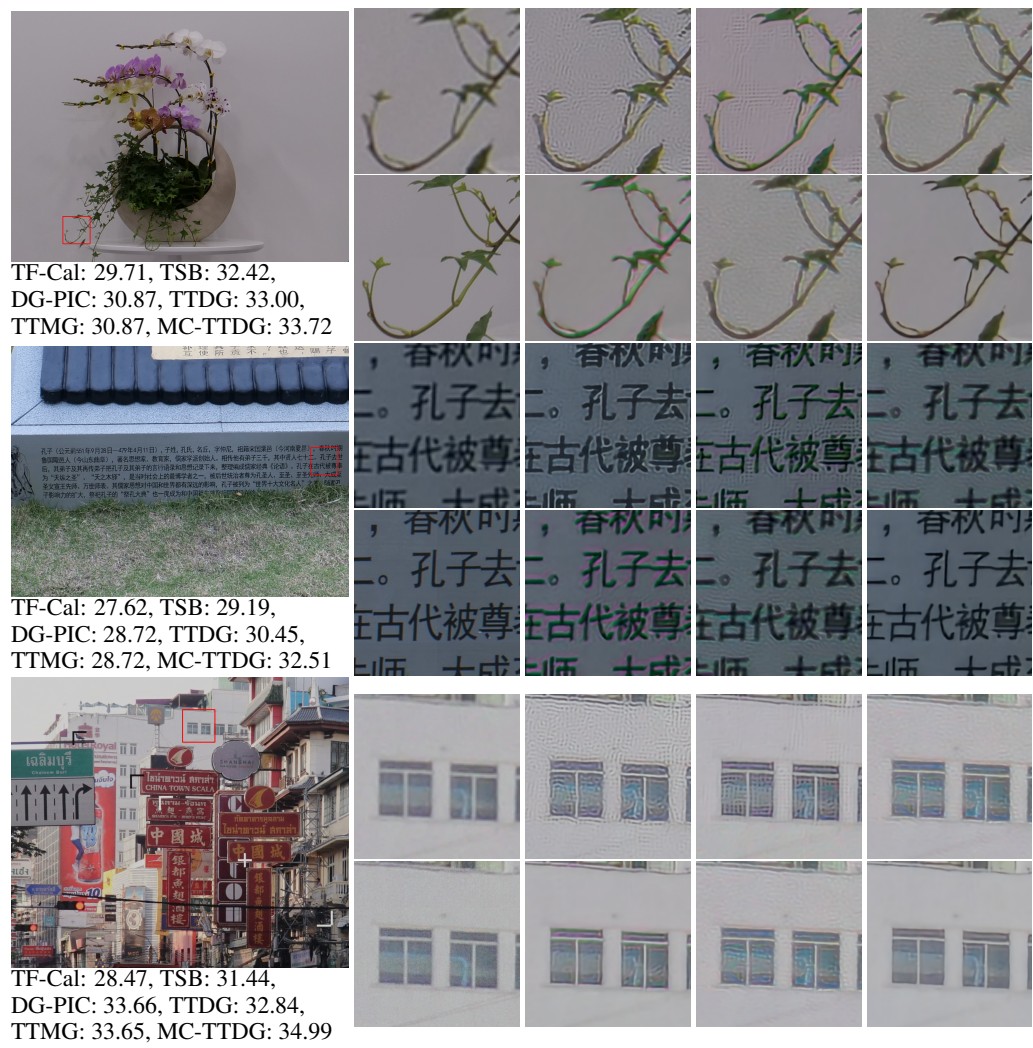

Figure 6: Part 1 of the visualization display diagram. The large image on the left is the LR image, and the sub-images on the right are LR, TF-Cal(Zhao et al., 2022), TSB(Park et al., 2023) and DG-PIC (Jiang et al., 2024)(first row), GT, TTDG(Zhou et al., 2024), and TTMG(Nam & Lee, 2025) and MC-TTDG (Ours) (second row). The values following the name represent the PSNR metric of the current patch. Please zoom-in on screen. Please zoom-in on screen.

experiments, analyze results, or contribute to the research methodology. All scientific content, including the conceptualization, design, implementation, and validation of the work, was entirely carried out by the authors.

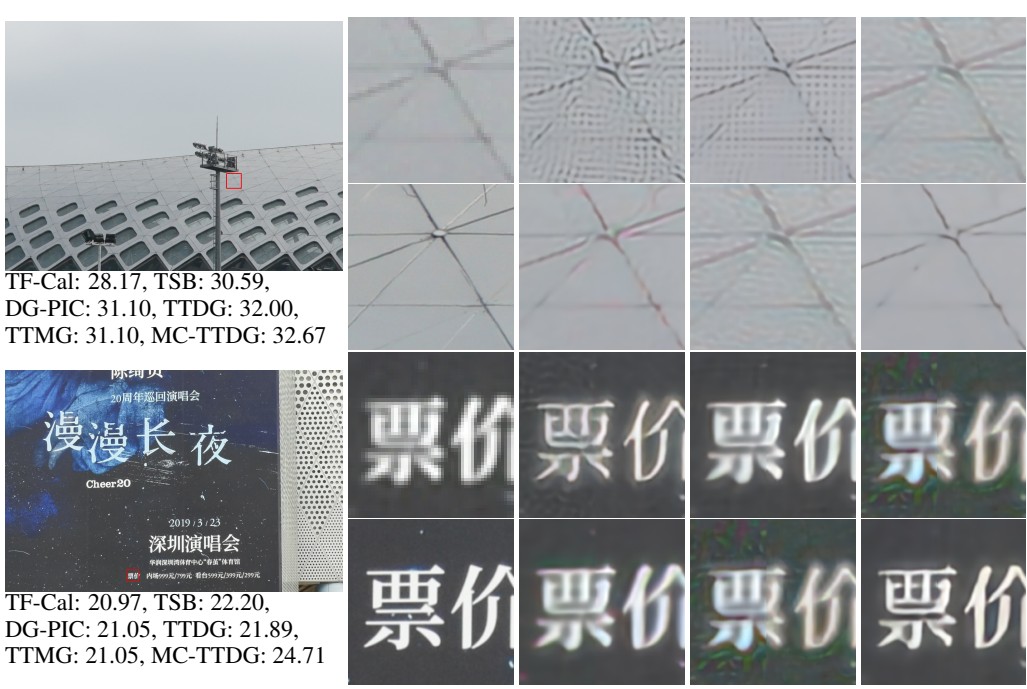

Figure 7: Part 2 of the visualization display diagram. The large image on the left is the LR image, and the sub-images on the right are LR, TF-Cal(Zhao et al., 2022), TSB(Park et al., 2023) and DG-PIC (Jiang et al., 2024)(first row), GT, TTDG(Zhou et al., 2024), and TTMG(Nam & Lee, 2025) and MC-TTDG (Ours) (second row). The values following the name represent the PSNR metric of the current patch. Please zoom-in on screen. Please zoom-in on screen.

