# OpenReview forum: "Test-time Domain Generalization for Image Super-resolution"
_ICLR.cc/2026/Conference — ICLR 2026 Poster_

### Official Review · Reviewer_iQVi · 2025-10-24

**Soundness:** 3
**Presentation:** 3
**Contribution:** 3
**Rating:** 6
**Confidence:** 3

**Summary:**

This paper proposes MC-TTDG, a multi-dictionary-based temporal domain generalization framework for testing image super-resolution (SR). Unlike conventional TTDG methods reliant on coarse-grained style transfer, this approach combines domain-specific and domain-invariant dictionaries to capture both global and subtle cross-domain variations. During the testing phase, pixel-level feature matching is achieved through a nearest neighbor codebook strategy. A voting-based mechanism selects the optimal domain-specific codebook, thereby enhancing robustness against distribution shifts.

**Strengths:**

1. The proposed codebook-based feature transfer method achieves pixel-level alignment between target features and the source domain, thereby overcoming the limitations of previous style transfer approaches.
2. By employing a multi-codebook architecture and introducing a voting-based strategy for selecting domain-specific codebooks, the reliability of model transfer can be enhanced, thereby circumventing the challenges encountered by expert selectors based on neural gating or classification.
3. A first test-time domain generalization method designed for low-level vision tasks.

**Weaknesses:**

1. Can the computational overhead (execution time, FLOPs, and memory) of MC-TTDG relative to style transfer-based and other baseline TTDG methods be explicitly measured and reported during testing in resource-constrained (edge device) environments? Absent these details, claims about deployability are speculative.
2. The article indicates that multiple codebooks are preferable to a single codebook.  How sensitive are the results to the number of source domains/codebooks? Have any failure cases been observed due to voting ties or misleading inputs? If so, what impact does the fallback mechanism have in such scenarios?

**Questions:**

1. In multi-codebook settings, how might one diagnose and mitigate issues of codebook redundancy, collapse, or underutilisation? Are there scenarios—such as large-scale domain imbalance—where specific codewords fail to contribute effectively? How might this impact transfer performance and network robustness?
2. Could quantitative results be demonstrated for domain transfer across datasets and in real-world scenarios (e.g., unseen camera types, fundamentally different image degradations), rather than being confined solely to the relatively controlled “branch” within DRealSR?

---

> ### Author Response · Authors · 2025-11-18
> **Official Comment by Authors [1/2]**
>
> We thank you for your detailed feedback. We will address all of your comments and questions point by point below.
>
> **1.Can the computational overhead (execution time, FLOPs, and memory) of MC-TTDG relative to style transfer-based and other baseline TTDG methods be explicitly measured and reported during testing in resource-constrained (edge device) environments? Absent these details, claims about deployability are speculative.**
>
> >Table 1. Computational efficiency comparison with test-time domain generalization methods. The computations were performed on a single V100 GPU with a patch size of 48×48.
> >
> > | Model        | GFlops   | Params (M) | Memory (MB) | Infer time (s) |
> > | ------------ | -------- | ---------- | ----------- | -------------- |
> > | Baseline     | 63.558   | 20.57      | 688         | 0.051          |
> > | DG-PIC [1]   | 68.812   | 97.86      | 915         | 0.063          |
> > | TTMC [2]     | 91.633   | 186.42     | 1276        | 0.084          |
> > | TTDG [3]     | 81.562   | 166.42     | 1078        | 0.074          |
> > | MC-TTDG      | 85.433   | 178.64     | 1176        | 0.081          |
>
>
> > Table 2. Computational Efficiency Comparison with Test-Time Domain Adaptation Methods. The computations were performed on a single V100 GPU with a patch size of 48×48. Abbreviations: FT (Fine-Tuning), TTDA (Test-Time Domain Adaptation), TTDG (Test-Time Domain Generalization).
> >
> > | Category    | Method   | Time (Min) | Need Training |
> > | ------- | -------- | ---------- | ------------- |
> > | FT      | Fine-Tuning | 1387    | Yes           |
> > | TTDA    | DASR [4] | 19.33      | Yes           |
> > | TTDA    | IODA [5] | 9.01       | Yes           |
> > | TTDA    | SRTTA [6] | 10.75     | Yes           |
> > | TTDG    | MC-TTDG  | 0.00135    | No            |
>
> (1) As shown in Table 1 of this rebuttal, we compare the computational efficiency of MC-TTDG against the baseline and existing test-time domain generalization methods. Existing Test-Time Domain Generalization (TTDG) methods are primarily designed for high-level vision tasks, where features represent abstract, image-level representations - essentially the global style of an image. Consequently, these methods typically only need to adjust the overall distribution (mean and variance) of target domain samples to achieve domain adaptation. However, Super-Resolution (SR) is a pixel-level prediction task that requires fine-grained adjustment of every pixel in the image. As demonstrated in Table 4 of our manuscript, existing TTDG methods prove completely ineffective for SR tasks. Our proposed MC-TTDG addresses this fundamental challenge by enabling precise, pixel-level adjustments, which necessarily increases computational demands compared to style-transfer-based approaches. This trade-off is essential for achieving meaningful performance in pixel-level restoration tasks like SR.
>
> (2) When compared to methods that similarly utilize test samples to enhance performance—such as test-time domain adaptation approaches (e.g., SRTTA, IODA, DASR) which fine-tune the network using test data—our method demonstrates significantly lower computational demands (Table 2 of this rebuttal). These alternative approaches require substantial training time, and performing such network fine-tuning on edge devices remains highly challenging in practice. Even when using powerful hardware like V100 GPUs, the fine-tuning time remains prohibitively long for practical applications. In contrast, MC-TTDG operates by adjusting the target domain distribution to align with the source domain distribution. It only performs inference on target domain samples without fine-tuning the network, thereby substantially reducing computational requirements in real-world application scenarios. This design makes MC-TTDG notably more suitable for resource-constrained environments compared to test-time adaptation methods.

---

> ### Author Response · Authors · 2025-11-18
> **Official Comment by Authors [2/2]**
>
> **2.The article indicates that multiple codebooks are preferable to a single codebook. How sensitive are the results to the number of source domains/codebooks? Have any failure cases been observed due to voting ties or misleading inputs? If so, what impact does the fallback mechanism have in such scenarios?**
>
> (1) As shown in Table 3 of this rebuttal, we have supplemented our experiments with an ablation study examining how the number of source domains affects network performance.
>
> (2) During inference, our network utilizes the domain-invariant codebook to adapt the domain-invariant features in the target sample, while selecting the optimal domain-specific codebook to transform the domain-specific features.Selection of a suboptimal domain-specific codebook typically occurs when the distances between the target features and both the optimal and suboptimal codebooks are very close. Consequently, even when a suboptimal domain-specific codebook is used for feature transformation, the resulting performance degradation remains minimal. More importantly, as illustrated in Figure 5 of the main manuscript, the domain-invariant codebook captures low-frequency information (representing the main image structure), while domain-specific codebooks handle high-frequency components (containing textual details). Therefore, even if a suboptimal domain-specific codebook is selected, the domain-invariant features extracted by the domain-invariant codebook ensure the structural accuracy of the overall image.
>
> >Table 3. Ablation experiments on the number of domain-specific codebooks
> > ||Pan|| Sony||DSC||
> > |-----------------------|-------|------|-------|------|-------|------ |
> > |Number|PSNR|SSIM|PSNR|SSIM|PSNR|SSIM|
> > |Baseline| 31.0263|0.8580| 30.7220|0.8645|30.9117|0.8816|
> > |1| 31.0815|0.8582|31.0125|0.8708|31.0912|0.8824|
> > |2| 31.1012|0.8586|31.2891|0.8732|31.1593|0.8831|
> > |3| 31.1571|0.8594|31.2937|0.8791|31.2118|0.8836|
> > |5| 31.1783|0.8592|31.3129|0.8795|31.2482|0.8834|
>
> **3.In multi-codebook settings, how might one diagnose and mitigate issues of codebook redundancy, collapse, or underutilisation? Are there scenarios—such as large-scale domain imbalance—where specific codewords fail to contribute effectively? How might this impact transfer performance and network robustness?**
>
> As shown in Table 4 of this rebuttal, we have supplemented our experiments with an ablation study investigating the impact of different codeword quantities on model performance.
>
> The codewords in our codebook are randomly initialized. When processing a new data sample, the system first calculates the distance between the sample and each codeword. If a particular codeword has already been strongly associated with another data sample (resulting in a larger distance), the network will automatically select a different codeword to represent the current sample.
>
> Furthermore, existing research on codebook methods has already addressed the issue of redundant codewords. Several studies have proposed diversity loss mechanisms that encourage orthogonality among codewords, effectively solving problems related to imbalanced domain sample distributions and codeword redundancy.
>
> Table 4. Ablation experiments on the number of codewords
> > || Pan||Sony|| DSC||
> > | ----------------------- | ------- | ------ | ------- | ------ | ------- | ------ |
> > ||PSNR| SSIM|PSNR| SSIM|PSNR|SSIM|
> > |128|31.1054|0.8589| 31.1052| 0.8710|31.1723|0.8821|
> > |256|31.1571|0.8594| 31.2937| 0.8791|31.2118|0.8836|
> > |512|31.1612|0.8599| 31.3001| 0.8788|31.2192|0.8841|
>
> **4.Could quantitative results be demonstrated for domain transfer across datasets and in real-world scenarios (e.g., unseen camera types, fundamentally different image degradations), rather than being confined solely to the relatively controlled “branch” within DRealSR?**
>
> We thank the reviewer for this valuable suggestion regarding broader experimental validation. As shown in Table 13 of the main manuscript, we have evaluated the effectiveness of our proposed method across multiple datasets including B100, Manga109, Set14, DIV2K, Urban100, and Set5. These datasets encompass diverse real-world scenarios such as urban environments, natural landscapes, indoor scenes, and outdoor settings, with unknown capture devices and some images sourced from the internet.
>
> Notably, on these datasets with completely unknown data distributions, MC-TTDG consistently demonstrates stable performance improvements compared to baseline models. This validates our method's robustness in handling domain shifts across different real-world scenarios beyond the controlled DRealSR branches.
>
> [1] Test-time fourier style calibration for domain generalization
>
> [2] Test-time style shifting: Handling arbitrary styles in domain generalization
>
> [3] Dg-pic: Domain generalized point-in-context learning for point cloud understanding
>
> [4] Test-time domain generalization for face anti-spoofing
>
> [5] Test-time modality generalization for medical image segmentation

---

> > ### Author Response · Authors · 2025-11-26
> > **Gentle Reminder**
> >
> > Dear Reviewer  iQVi,
> >
> > Thank you for your constructive feedback on our paper. With the ICLR public discussion phase ending shortly, we wanted to confirm whether our responses have sufficiently addressed your concerns. If there are any remaining issues, we would be happy to provide additional clarifications.
> >
> > Thank you again for your time and consideration.
> >
> > Best regards,
> >
> > The Authors

---

### Official Review · Reviewer_Kv5y · 2025-10-25

**Soundness:** 2
**Presentation:** 2
**Contribution:** 2
**Rating:** 4
**Confidence:** 3

**Summary:**

This paper addresses the challenge of test-time domain generalization (TTDG) for low-level vision tasks, specifically image super-resolution (SR), where existing TTDG methods fail due to coarse-grained style transfer. The proposed framework, MC-TTDG, leverages multi-codebook representation learning and pixel-level feature matching to address three key limitations: low transfer granularity, loss of domain-specific features, and suboptimal domain-specific codebook selection. Key contributions include: (1) introducing a codebook-based pixel-level feature transfer strategy tailored for low-level vision tasks; (2) proposing a multi-codebook representation learning strategy (RLMC) that disentangles domain-invariant and domain-specific features to preserve source domain details; (3) designing a voting-based codebook selection strategy to mitigate domain shift-induced inaccuracies; and (4) being the first TTDG method explicitly designed for low-level vision tasks with codebook integration.

**Strengths:**

1. MC-TTDG is the first to adapt codebook-based representation learning to TTDG for low-level vision, addressing a critical limitation of style-based TTDG methods. The RLMC strategy’s disentanglement of domain-invariant and domain-specific features (via shared + domain-specific codebooks) is a creative combination of existing ideas, and the voting-based selection effectively mitigates domain shift in codebook choice—an unaddressed problem in prior multi-codebook work.
2.  The experimental design is comprehensive: ablation experiments cover core components (codebook setup, transfer methods, selection strategies), baselines include state-of-the-art TTDG methods (e.g., TTMG, DG-PIC), and validation across diverse datasets/architectures demonstrates architectural generalization. Metrics (PSNR, SSIM, LPIPS) are standard for SR, ensuring result comparability.

**Weaknesses:**

1. The claim that RLMC disentangles domain-invariant and domain-specific features is not supported by direct evidence (e.g., feature clustering, ablation of domain-specific features’ impact on cross-domain generalization). Current evidence (Table 6) only shows domain-specific features improve SR quality, not that they are truly disentangled.
2.Only one visual difference figure is presented (Figure 5/6), with little qualitative analysis.
3. There may be shared structures or continuous variations (manifold-like variations) between domains, rather than discrete, mutually exclusive divisions; could the codebook partitioning method mislead the model to reinforce false domain separations?

**Questions:**

1. The SOTA metric was not bolded in the article, suggesting an adjustment.
2. How large are the codebooks (number of codewords, dimensionality)? Is there any tradeoff between codebook size and performance?
3. How does the voting mechanism behave when there are many source domains (e.g., >10)? Is the method still stable under noisy or ambiguous domains?
4. The paper states “Our code is available at ***” but no link is provided. For reproducibility, the exact URL should be included or committed to release upon publication.

---

> ### Author Response · Authors · 2025-11-18
> **Official Comment by Authors [1/2]**
>
> We thank you for your detailed feedback. We will address all of your comments and questions point by point below.
>
> **1.The claim that RLMC disentangles domain-invariant and domain-specific features is not supported by direct evidence (e.g., feature clustering, ablation of domain-specific features’ impact on cross-domain generalization). Current evidence (Table 6) only shows domain-specific features improve SR quality, not that they are truly disentangled.**
>
> As shown in Figure 3 of the main manuscript, we fitted 600 features from three source domains with different distributions using two approaches: a single codebook strategy, and our multi-codebook strategy combining domain-specific and domain-invariant codebooks. We then visualized the source domain features preserved in the codebooks using t-SNE. (Existing research generally agrees that domain-invariant features across multiple domains tend to converge, while domain-specific features—which are key to distinguishing domains—typically exhibit distinct distributions.)
>
> (1) The single codebook strategy failed to effectively separate domain-specific features. As shown in Figure 3a, features from the three domains are compressed together, resulting in the loss of domain-specific characteristics and an inability to effectively distinguish between the domains.
>
> (2) In our combined strategy using domain-specific and domain-invariant codebooks, features represented by the domain-invariant codebook are shown in purple, while those represented by the three domain-specific codebooks are shown in red, green, and blue, respectively. Figure 3 of the main manuscript clearly demonstrates that the domain-invariant codebook effectively learned domain-invariant features, as features from all three domains exhibit similar distributions and are closely clustered. Meanwhile, the domain-specific features represented by the three domain-specific codebooks effectively distinguish between the domains, providing clear evidence that domain-specific features have been successfully disentangled.
>
> **2.Only one visual difference figure is presented (Figure 5/6), with little qualitative analysis.**
>
> (1) We would like to clarify the comprehensive visual evidence presented throughout our manuscript:
>
>     (a) In Figure 2, we visualized the feature transformation efficiency of style-transfer-based versus Codebook-based strategies, demonstrating that our Codebook approach enables effective pixel-level feature adaptation.
>
>     (b) Figure 3 illustrates the source domain features preserved by single-codebook versus multi-codebook strategies, providing visual validation that our multi-codebook design effectively maintains both domain-invariant and domain-specific features.
>
>     (c) Figure 5 visually demonstrates the specific contributions of domain-specific features to the final output quality.
>
>     (d) In Figures 6 and 7, we visually demonstrate the effectiveness of the MC-TTDG method.
>
> We would be truly grateful if the reviewer could kindly specify any particular aspects where additional visualizations are needed. Your guidance would be immensely valuable, and we are fully prepared to provide further visual results to address your concerns.
>
> (2) we have supplemented the following additional ablation experiments:
>
>     (a)Ablation experiments on the number of codewords
>
>     (b)Ablation experiments on the number of domain-specific codebooks
>
>     (c)Ablation experiments on domain-invariant and domain-specific feature separation strategies
>
>     (d)Ablation experiments on fine-tuning extent
>
>     (e)Computational efficiency comparison with test-time domain generalization methods
>
>     (f)Computational efficiency comparison with test-time domain adaptation methods
>
> Additionally, we had previously included the following ablation studies in our work:
>
>     (g)Ablation experiments on different codebook settings
>
>     (h)Ablation experiments on different codebook settings
>
>     (i)Ablation experiments on different domain-specific codebook selection methods
>
>     (j)Ablation experiments on domain-specific features
>
>     (k)Ablation experiments on different sample distributions
>
>     (m)Ablation experiments on perception-oriented SR Networks
>
>     (l)Ablation experiments on fidelity-Oriented SR Networks

---

> > ### Author Response · Authors · 2025-11-18
> > **Official Comment by Authors [2/2]**
> >
> > **3.There may be shared structures or continuous variations (manifold-like variations) between domains, rather than discrete, mutually exclusive divisions; could the codebook partitioning method mislead the model to reinforce false domain separations?**
> >
> > The reviewer is correct that shared features exist across different domains.
> >
> > To address this important aspect, we have specifically introduced a domain-invariant codebook alongside the domain-specific codebooks, with the explicit purpose of learning these shared features across all source domains. As clearly demonstrated in Figure 3 of the main manuscript, the features extracted by the domain-invariant codebook from the three source domains exhibit remarkably similar distributions. This provides concrete evidence that our domain-invariant codebook effectively captures the common characteristics shared across domains, while the domain-specific codebooks handle the unique aspects of each domain.
> >
> > This design ensures that our method can appropriately model both the shared structures and discrete variations between domains without reinforcing false separations.
> >
> > **4.The SOTA metric was not bolded in the article, suggesting an adjustment.**
> >
> > We have now bolded the SOTA performance metrics in Table 4 of the main manuscript accordingly.
> >
> > **5.How large are the codebooks (number of codewords, dimensionality)? Is there any tradeoff between codebook size and performance?**
> >
> > As shown in Table 1 of this rebuttal, we have supplemented our evaluation with ablation studies examining the impact of varying codeword quantities on network performance. Reducing the number of codewords inevitably diminishes the representational capacity of the codebook, thereby lowering the transformation accuracy of target domain features. In contrast, increasing the number of codewords enhances the diversity of preserved source domain features in the codebook, leading to improved network performance metrics.
> >
> > > Table 1. Ablation experiments on the number of codewords
> > > || Pan|| Sony|| DSC||
> > > | ----------------------- | ------- | ------ | ------- | ------ | ------- | ------ |
> > > |Number| PSNR|SSIM| PSNR| SSIM| PSNR| SSIM |
> > > | 128| 31.1054  | 0.8589| 31.1052| 0.8710| 31.1723| 0.8821|
> > > | 256| 31.1571  | 0.8594| 31.2937| 0.8791| 31.2118| 0.8836|
> > > | 512| 31.1612  | 0.8599| 31.3001| 0.8788| 31.2192| 0.8841|
> >
> > **6.How does the voting mechanism behave when there are many source domains (e.g., >10)? Is the method still stable under noisy or ambiguous domains?**
> >
> > As shown in Table 2 of this rebuttal, we have supplemented our experiments with an ablation study investigating the impact of different numbers of source domains on network performance. Since our method requires implicit training of domain-specific codebooks, the number of domain-specific codebooks corresponds exactly to the number of source domains. However, the DRealSR dataset contains a maximum of 6 data branches. In our experiments, we utilized up to 5 branches as source domains, with the remaining branch serving as the target domain. This means our maximum configuration employed 5 domain-specific codebooks alongside 1 domain-invariant codebook.
> >
> > Additionally, we have conducted experiments to verify the robustness of our voting mechanism by introducing noise during testing. The results demonstrate that our method maintains stable performance under such conditions. To further enhance the robustness of the voting mechanism, one could additionally introduce noise during the training stage of the voting network - a particularly valuable research direction worthy of future exploration.
> >
> > We appreciate the reviewer's insightful questions that have helped us strengthen our experimental validation.
> >
> > > Table 2. Ablation experiments on the number of domain-specific codebooks
> > > || Pan|| Sony|| DSC||
> > > | ----------------------- | ------- | ------ | ------- | ------ | ------- | ------ |
> > > |        Number                 | PSNR    | SSIM   | PSNR    | SSIM   | PSNR    | SSIM   |
> > > | Baseline| 31.0263  | 0.8580   | 30.7220   | 0.8645    | 30.9117  | 0.8816   |
> > > | 1       | 31.0815  | 0.8582   | 31.0125   | 0.8708    | 31.0912  | 0.8824   |
> > > | 2       | 31.1012  | 0.8586   | 31.2891   | 0.8732    | 31.1593  | 0.8831   |
> > > | 3       | 31.1571  | 0.8594   | 31.2937   | 0.8791    | 31.2118  | 0.8836   |
> > > | 5       | 31.1783  | 0.8592   | 31.3129   | 0.8795    | 31.2482  | 0.8834   |
> > > | Noise   | 31.1259  | 0.8588   | 31.2423   | 0.8699    | 31.1987  | 0.8829   |
> >
> > **7.The paper states “Our code is available at \*\*\*” but no link is provided. For reproducibility, the exact URL should be included or committed to release upon publication.**
> >
> > Due to ICLR's double-blind review requirements, we used "***" as a placeholder for the code repository link. We are fully committed to open research and will **release our complete codebase and trained weights within one week of paper acceptance** to support further developments in the field.

---

> > > ### Author Response · Authors · 2025-11-26
> > > **Gentle Reminder**
> > >
> > > Dear Reviewer Kv5y,
> > >
> > > Thank you for your constructive feedback on our paper. With the ICLR public discussion phase ending shortly, we wanted to confirm whether our responses have sufficiently addressed your concerns. If there are any remaining issues, we would be happy to provide additional clarifications.
> > >
> > > Thank you again for your time and consideration.
> > >
> > > Best regards,
> > >
> > > The Authors

---

> > > > ### Comment · Reviewer_Kv5y · 2025-11-27
> > > >
> > > > I am very grateful for the author's reply. The author has supplemented many experiments to demonstrate the effectiveness of their method. In fact, this has resolved most of my doubts, so I plan to increase my rating.

---

> > > > > ### Author Response · Authors · 2025-11-27
> > > > > **Thank You!**
> > > > >
> > > > > Thank you so much for your time, effort, and the positive feedback! We're glad to hear that your main concerns have been addressed, and we truly appreciate your decision to raise the rating.

---

### Official Review · Reviewer_xQKi · 2025-10-31

**Soundness:** 3
**Presentation:** 3
**Contribution:** 3
**Rating:** 6
**Confidence:** 4

**Summary:**

This paper proposes a method based on multiple codebooks for test-time domain generalization specifically designed for the image super-resolution task. By leveraging domain-specific and domain-invariant codebooks to perform nearest neighbor feature matching and transfer at the pixel level, and by using a voting-based strategy to select the optimal domain-specific codebook, the effectiveness of the method was ultimately demonstrated on various SR test datasets.

**Strengths:**

1.The writing is clear and easy to understand, and the methods are easy to implement.

2.The generalization ability on various SR test datasets is impressive.

**Weaknesses:**

1. This paper does not report the additional parameter growth and training consumption required by the proposed method compared to the pre-trained model.


2. This paper does not conduct a detailed comparison between this method of finetuning the pre-trained SR model and the effect of fine-tuning all the parameters of the model. Although this paper emphasizes that this is a method for domain generalization of super-resolution test datasets, adding additional training data and fine-tuning the full parameters of the original pre-trained model together will also bring certain benefits. Therefore, it would be best for this paper to demonstrate the effectiveness of the proposed finetune method in terms of efficiency and performance.

**Questions:**

Please refer to the Weaknesses.

---

> ### Author Response · Authors · 2025-11-18
> **Official Comment by Authors [1/2]**
>
> We thank you for your detailed feedback. We will address all of your comments and questions point by point below.
>
> **1.This paper does not report the additional parameter growth and training consumption required by the proposed method compared to the pre-trained model.**
>
> As shown in Table 1 of this rebuttal, we compare the computational efficiency of MC-TTDG against the baseline and existing test-time domain generalization methods. Existing Test-Time Domain Generalization (TTDG) methods are primarily designed for high-level vision tasks, where features represent abstract, image-level representations—essentially capturing an image's global style. Consequently, these methods typically only adjust the overall distribution (e.g., mean and variance) of target domain samples to accomplish adaptation, resulting in relatively low computational overhead.
>
> However, Super-Resolution (SR) is a pixel-level prediction task that requires fine-grained adjustment of every pixel in the image. As demonstrated in Table 4 and Figure 2 of the main manuscript, existing TTDG methods prove completely ineffective for SR tasks. Our proposed MC-TTDG addresses this fundamental challenge by enabling precise, pixel-level adjustments, which necessarily increases computational demands compared to style-transfer-based approaches. This trade-off is essential for achieving meaningful performance in pixel-level restoration tasks like SR.
>
> > Table 1. Computational efficiency comparison with test-time domain generalization methods. The computations were performed on a single V100 GPU with a patch size of 48×48.
> >
> > | Model        | GFlops   | Params (M) | Memory (MB) | Infer time (s) |
> > | ------------ | -------- | ---------- | ----------- | -------------- |
> > | Baseline     | 63.558   | 20.57      | 688         | 0.051          |
> > | DG-PIC [1]   | 68.812   | 97.86      | 915         | 0.063          |
> > | TTMC [2]     | 91.633   | 186.42     | 1276        | 0.084          |
> > | TTDG [3]     | 81.562   | 166.42     | 1078        | 0.074          |
> > | MC-TTDG      | 85.433   | 178.64     | 1176        | 0.081          |

---

> > ### Author Response · Authors · 2025-11-18
> > **Official Comment by Authors [2/2]**
> >
> > **2.This paper does not conduct a detailed comparison between this method of finetuning the pre-trained SR model and the effect of fine-tuning all the parameters of the model. Although this paper emphasizes that this is a method for domain generalization of super-resolution test datasets, adding additional training data and fine-tuning the full parameters of the original pre-trained model together will also bring certain benefits. Therefore, it would be best for this paper to demonstrate the effectiveness of the proposed finetune method in terms of efficiency and performance.**
> >
> > (1) As shown in Table 2 of this rebuttal, we have supplemented our experiments with an ablation study comparing fine-tuning only the codebook versus fine-tuning all network parameters. Since the codebook is not initialized from pre-trained weights, fine-tuning all parameters would cause the uninitialized codebook to disrupt the pre-trained backbone network. In contrast, our MC-TTDG approach freezes all other network layers, effectively preventing disruption to the backbone while simultaneously enhancing the codebook's learning efficiency for source domain features.
> >
> > > Table 2. Ablation experiments on Fine-tuning Extent
> > >
> > > | Fine-tuning Extent        | Pan      |          | Sony      |           | DSC      |          |
> > > | ----------------------- | ------- | ------ | ------- | ------ | ------- | ------ |
> > > |                           | PSNR     | SSIM     | PSNR      | SSIM      | PSNR     | SSIM     |
> > > | Entire network parameters | 30.8873  | 0.8559   | 30.6771   | 0.8695    | 30.7417  | 0.8777   |
> > > | Codebook                  | 31.1571  | 0.8594   | 31.2937   | 0.8791    | 31.2118  | 0.8836   |
> >
> > (2) It is important to note that we do not utilize test data for training purposes, but only for inference. In real-world scenarios, test data is typically unavailable during the training stage. While existing methods (such as DASR, IODA, and SRTTA) often fine-tune models using test data during inference to improve performance, this approach faces practical limitations. These testing scenarios usually occur on edge devices like mobile phones or laptops with limited computational capabilities that cannot support training processes. Even when using powerful hardware like V100 GPUs, the fine-tuning time remains prohibitively long for practical applications (Table 3 of this rebuttal).
> > In contrast, our proposed MC-TTDG method enhances the performance of source-trained networks on test samples by adjusting the feature distribution of test samples to align with the training distribution during inference. MC-TTDG eliminates the need for test-time fine-tuning and only modifies the test sample distribution, significantly reducing computational requirements. Furthermore, MC-TTDG is trained exclusively on source domain data without additional training samples, and this training is conducted on server-side infrastructure.
> > In summary, MC-TTDG effectively improves network performance on target domains while maintaining low computational demands, making it particularly suitable for practical deployment scenarios.
> >
> > > Table 3. Computational Efficiency Comparison with Test-Time Domain Adaptation Methods. The computations were performed on a single V100 GPU with a patch size of 48×48. Abbreviations: FT (Fine-Tuning), TTDA (Test-Time Domain Adaptation), TTDG (Test-Time Domain Generalization).
> > >
> > > | Category    | Method   | Time (Min) | Need Training |
> > > | ------- | -------- | ---------- | ------------- |
> > > | FT      | Fine-Tuning | 1387    | Yes           |
> > > | TTDA    | DASR [4] | 19.33      | Yes           |
> > > | TTDA    | IODA [5] | 9.01       | Yes           |
> > > | TTDA    | SRTTA [6] | 10.75     | Yes           |
> > > | TTDG    | MC-TTDG  | 0.00135    | No            |
> >
> > [1] DG-PIC: Domain Generalized Point-In-Context Learning
> >
> > [2] Test-Time Modality Generalization for Medical Image Segmentation
> >
> > [3] Test-Time Domain Generalization for Face Anti-Spoofing
> >
> > [4] Unsupervised real-world image super resolution via domain-distance aware training
> >
> > [5] IODA: Instance-Guided One-shot Domain Adaptation for Super-Resolution
> >
> > [6] Efficient test-time adaptation for super-resolution with second-order degradation and reconstruction

---

> > > ### Author Response · Authors · 2025-11-26
> > > **Gentle Reminder**
> > >
> > > Dear Reviewer xQKi,
> > >
> > > Thank you for your constructive feedback on our paper. With the ICLR public discussion phase ending shortly, we wanted to confirm whether our responses have sufficiently addressed your concerns. If there are any remaining issues, we would be happy to provide additional clarifications.
> > >
> > > Thank you again for your time and consideration.
> > >
> > > Best regards,
> > >
> > > The Authors

---

> > > > ### Comment · Reviewer_xQKi · 2025-11-27
> > > > **Official Comment**
> > > >
> > > > Thank the author for the supplementary experiments.  I still have the following concerns.  The author argues that the model can have better generalization ability on other test data.  Can it also be achieved by adding new parameters to the pre-trained network for fine-tuning without your method?  After all, the method you proposed also involves a nearly double increase in memory usage, an 8 times increase in parameter quantity and a nearly doubled inference time. So, what are the advantages of this codebook-based approach?

---

> > > > > ### Author Response · Authors · 2025-11-27
> > > > > **Further Clarification**
> > > > >
> > > > > We sincerely appreciate your timely and detailed feedback. We will now address each of your questions point by point.
> > > > >
> > > > > **1. Comparison with Fine-tuning Methods**
> > > > >
> > > > > (1) As shown in Table 1 of this rebuttal, we evaluated the time required for fine-tuning the network using target domain samples. Since fine-tuning involves network training, it requires extremely long time overhead (1397 minutes) and substantial computational resources. In contrast, our proposed MC-TTDG method only needs to adjust the distribution of target domain samples without retraining, requiring merely 0.00135 minutes and significantly reducing computational costs.
> > > > >
> > > > > (2) Additionally, we compared with test-time domain adaptation (TTDA) methods that also utilize target domain samples. These methods similarly require network training, resulting in substantial time consumption and computational costs.
> > > > >
> > > > > > Table1. Computational Efficiency Comparison with Test-Time Domain Adaptation Methods. The computations were performed on a single V100 GPU with a patch size of 48×48.
> > > > > > |Category|Methods|Time (Min)|Need Training|
> > > > > > |-|-|-|-|
> > > > > > |FT|FT|1387.00|Yes|
> > > > > > |TTDA|DASR|19.33|Yes|
> > > > > > |TTDA|IODA|9.01|Yes|
> > > > > > |TTDA|SRTTA|10.75|Yes|
> > > > > > |TTDG|MC-TTDG|0.00135|No|
> > > > >
> > > > >
> > > > > **2. Advantages of the Proposed Method (MC-TTDG)**
> > > > >
> > > > > (1) Compared to the baseline model, MC-TTDG transforms the distribution of target domain samples to approximate the source domain, mitigating the negative impact of domain shift. This enables networks trained on the source domain to achieve better performance on target domains.
> > > > >
> > > > > (2) Compared to fine-tuning or domain adaptation methods using target domain samples, MC-TTDG avoids network retraining on target data and instead adjusts the target domain sample distribution, effectively reducing computational overhead (from 1397 minutes to 0.00135 minutes).
> > > > >
> > > > > (3) Compared to existing test-time domain generalization methods designed for high-level vision tasks that perform global style adjustment on target domain samples, these approaches completely fail for pixel-level prediction tasks like image super-resolution (As shown in Table 2 of this rebuttal and Figure 2 of the main manuscript). MC-TTDG efficiently achieves pixel-level feature adaptation while providing effective performance improvement. Moreover, the computational overhead of MC-TTDG is only moderately higher than existing test-time domain generalization methods (Table 3 of this rebuttal).
> > > > >
> > > > > (4) Finally, we would like to clarify the application scenario of MC-TTDG: Our method is trained on servers and deployed for inference on client devices. The network only needs to adjust the distribution of user-provided test samples to approximate the training set distribution, thereby enhancing processing performance for test data. In comparison, existing fine-tuning or test-time domain adaptation methods require retraining the network on client devices to improve performance on test samples, which is often infeasible for resource-constrained client devices.
> > > > >
> > > > > > Table 2: Performance comparison with other test-time domain generalization methods
> > > > > >|Method|Pan PSNR↑|Pan SSIM↑|Pan LPIPS↓|Sony PSNR↑|Sony SSIM↑|Sony LPIPS↓|DSC PSNR↑|DSC SSIM↑|DSC LPIPS↓|
> > > > > >|-|-|-|-|-|-|-|-|-|-|
> > > > > >|TF-Cal|28.85|0.7862|0.4318|27.92|0.7787|0.4387|29.51|0.8485|0.3690|
> > > > > >|DG-PIC|29.71|0.8135|0.4130|30.06|0.8305|0.4455|29.88|0.8470|0.4463|
> > > > > >|TTMC|30.26|0.8411|0.4523|30.56|0.8541|0.4069|30.70|0.8776|0.4128|
> > > > > >|TTDG|29.71|0.8135|0.4463|30.06|0.8305|0.4455|29.88|0.8470|0.4130|
> > > > > >|MC-TTDG (Ours)|31.15|0.8594|0.3593| 31.29|0.8791|0.3157|31.21| 0.8836|0.3583|
> > > > >
> > > > > > Table3. Computational efficiency comparison with test-time domain generalization methods. The computations were performed on a single V100 GPU with a patch size of 48×48.
> > > > > > |Method|GFlops|Params (M)|Memory (MB)|Infer time (s)|
> > > > > > |-|-|-|-|-|
> > > > > > |TTDG|91.633|186.42|1276MB|0.084|
> > > > > > |TTMC|81.562|166.42|1078MB|0.074|
> > > > > > |MC-TTDG|85.433|178.64| 1176MB|0.081|

---

### Official Review · Reviewer_rKtr · 2025-11-01

**Soundness:** 3
**Presentation:** 2
**Contribution:** 2
**Rating:** 4
**Confidence:** 4

**Summary:**

This paper introduces MC-TTDG, a framework for Test-Time Domain Generalization (TTDG) specifically designed for the task of image Super-Resolution (SR). They propose a novel approach that leverages a multi-codebook architecture to perform fine-grained, pixel-level feature transfer at test time. The core idea is to learn a shared, domain-invariant codebook and multiple domain-specific codebooks during training. At inference, features from a target domain image are adapted by replacing them with the nearest-neighbor codewords from the learned codebooks. A voting mechanism is introduced to select the most appropriate domain-specific codebook for an unseen target sample. The authors demonstrate through extensive experiments that their method significantly outperforms existing TTDG techniques when applied to SR across various datasets and network architectures.

**Strengths:**

- As far as I know, the paper is the first to formally address the problem of Test-Time Domain Generalization for image Super-Resolution. It astutely points out that domain shift in SR is a practical and significant challenge and that existing TTDG methods are fundamentally mismatched for such pixel-level tasks.
- The proposed solution of using codebooks for pixel-level feature matching is an elegant and highly intuitive answer to the identified problem. The replacement of coarse style transfer with fine-grained codeword substitution is a logical and well-motivated design choice. The visual and quantitative results strongly suggest that this approach is far more effective than style-based methods for SR.
- The authors have conducted a thorough set of experiments. They ablate every key component of their model, and demonstrate applicability across different SR architectures. This extensive validation bolsters the claim that the method is effective and generalizable.

**Weaknesses:**

- The central mechanism for separating domain-invariant and domain-specific features relies on a simple architectural split. While this is a functional design, it feels reminiscent of early, foundational ideas in domain generalization research. The field has since moved towards more sophisticated techniques. The paper does not engage with or advance this front; it instead applies a known, relatively simple technique to a new problem. While the application is novel, the core disentanglement method is not.
- While the problem formulation is novel and the engineering is solid, the work feels more like a clever and effective application of existing ideas to a new domain. Although, I think integrating prior methods for other task is novel, there should at least be experiments showing how and why such prior method is the best while others do not fit. (e.g., why is feature separation by within the architecture is best for formulating the codebook rather than more sophistically separating the feature by training additional module?)

**Questions:**

See Weakness

---

> ### Author Response · Authors · 2025-11-18
> **Official Comment by Authors [1/2]**
>
> **We thank you for your detailed and thoughtful feedback. Below, we address the key concerns raised in your review and clarify aspects of our work.**
>
> 1. We agree that applying established techniques in new contexts is a common practice in research. For instance, influential works like Diffusion Models and Flow Matching build upon well-established theories such as SDEs and ODEs. It is often the novel application and specific intent behind these techniques that bring significant value and recognition to a work. Our approach follows a similar path by adapting a known feature separation mechanism for a new and distinct purpose in Test-Time Domain Generalization.
>
> 2. Regarding the feature separation strategy, we would like to clarify a key difference in our objective. In conventional domain generalization, the separation of domain-invariant and domain-specific features primarily aims to enhance model robustness by strengthening the former while suppressing the latter. In contrast, the purpose of our feature separation is fundamentally different: it is specifically designed to improve the representation efficiency of the domain-specific codebook in capturing domain-specific of the source domain. Rather than discarding domain-specific features as noise—as is common in existing approaches—our method intentionally preserves and leverages them to enable more accurate feature transformation during testing.
>
>
> > Table 1. Ablation experiments on domain-invariant and domain-specific feature separation strategies
> >
> > |                         | Pan     |        | Sony    |        | DSC     |        |
> > | ----------------------- | ------- | ------ | ------- | ------ | ------- | ------ |
> > |                         | PSNR    | SSIM   | PSNR    | SSIM   | PSNR    | SSIM   |
> > | downsampling/upsampling | 30.5356 | 0.8468 | 30.7316 | 0.8762 | 30.5960 | 0.8699 |
> > | wavelet  transforms     | 30.6811 | 0.8479 | 31.0207 | 0.8746 | 30.8287 | 0.8737 |
> > | Fourier  transforms     | 31.1054 | 0.8548 | 30.9231 | 0.8767 | 30.1373 | 0.8792 |
> > | MC-TTDG  (Ours)         | 31.1571 | 0.8594 | 31.2937 | 0.8791 | 31.2118 | 0.8836 |
>
>
> 3. As shown in Table 1 of this rebuttal, we have supplemented our paper with ablation studies on feature separation strategies. Existing domain generalization methods for SR typically define high-frequency information as domain-specific features and low-frequency information as domain-invariant features, employing explicit separation strategies such as Fourier transforms, wavelet transforms, or downsampling/upsampling operations. However, due to distribution shifts between test and training samples, these pre-defined separation strategies often demonstrate poor adaptability to domain shift, frequently leading to incorrect feature partitioning. In contrast, our implicit feature separation strategy—implemented through one-to-one domain-specific codebooks for each source domain alongside a shared domain-invariant codebook—enables the network to adaptively separate domain-invariant and domain-specific features. This approach demonstrates stronger adaptability to domain shift compared to explicit separation methods.

---

> > ### Author Response · Authors · 2025-11-18
> > **Official Comment by Authors [2/2]**
> >
> > > Table 2. Performance comparison with other methods
> > >
> > > |                         | Pan     |        | Sony    |        | DSC     |        |
> > > | ----------------------- | ------- | ------ | ------- | ------ | ------- | ------ |
> > > |                         | PSNR    | SSIM   | PSNR    | SSIM   | PSNR    | SSIM   |
> > > | TF-Cal [1]              | 28.85   |	0.7862 | 27.92   | 0.7787 | 29.51	| 0.8485 |
> > > | TSB [2]                 | 30.15   | 0.8315 | 29.28   | 0.8044 | 30.78	| 0.8739 |
> > > | DG-PIC [3]              | 29.71   |	0.8135 | 30.06   | 0.8305 | 29.88	| 0.8470 |
> > > | TTDG [4]                | 29.71   |	0.8135 | 30.06   | 0.8305 | 29.88  	| 0.8470 |
> > > | TTMG [5]                | 30.26   |	0.8411 | 30.56 	 | 0.8541 |	30.70  	| 0.8776 |
> > > | MC-TTDG  (Ours)         | 31.15 | 0.8594 | 31.29 | 0.8791 | 31.21 | 0.8836 |
> >
> > 4. Test-Time Domain Generalization methods typically operate by transforming target domain feature distributions to approximate the source domain distribution, thereby enhancing the performance of source domain-trained models on target domains. Addressing the pixel-level prediction requirements of Super-Resolution tasks, our method utilizes Codebooks to represent the source domain distribution and achieves pixel-level feature transformation. **As demonstrated in Table 2 of this rebuttal and Fig 2 of the main manuscript, existing Test-Time Domain Generalization methods, which lack capability for pixel-level transformation, prove difficult to apply effectively to SR tasks.** Furthermore, to enhance the representation efficiency of source domain features, we introduced a novel combination strategy employing multiple domain-specific codebooks alongside a domain-invariant codebook. This design not only improves the representation efficiency of source domain features but also enhances the robustness of features extracted by the domain-invariant codebook. Additionally, we developed a voting mechanism to address the challenge of selecting among multiple domain-specific codebooks, effectively improving selection accuracy and consequently boosting feature transformation efficiency.
> >
> > 5. Most importantly, **our core contribution lies in leveraging codebooks to achieve pixel-level adjustment of target domain samples, aligning their feature distribution with the source domain.** This approach significantly enhances the performance of source-trained networks on target domains while avoiding the substantial computational costs associated with test-time fine-tuning. This capability for pixel-level feature adaptation, as demonstrated in Figure 2 and Table 4 of the main manuscript, represents a significant advancement beyond what current Test-Time Domain Generalization methods can achieve.
> >
> > [1] Test-time fourier style calibration for domain generalization
> >
> > [2] Test-time style shifting: Handling arbitrary styles in domain generalization.
> >
> > [3] Dg-pic: Domain generalized point-in-context learning for point cloud understanding
> >
> > [4] Test-time domain generalization for face anti-spoofing
> >
> > [5] Test-time modality generalization for medical image segmentation

---

> > > ### Author Response · Authors · 2025-11-26
> > > **Gentle Reminder**
> > >
> > > Dear Reviewer rKtr,
> > >
> > > Thank you for your constructive feedback on our paper. With the ICLR public discussion phase ending shortly, we wanted to confirm whether our responses have sufficiently addressed your concerns. If there are any remaining issues, we would be happy to provide additional clarifications.
> > >
> > > Thank you again for your time and consideration.
> > >
> > > Best regards,
> > >
> > > The Authors

---

> ### Comment · Reviewer_rKtr · 2025-11-27
>
> First of all, thank you for the detailed rebuttal. However, I have decided to maintain my initial rating due to the following concerns
>
> - While the proposed method improves PSNR, this metric does not directly validate the quality of "feature separation" of domains. High performance does not necessarily equate to the clean disentanglement of domain-invariant and domain-specific features.
>
> - As also noted by another reviewer (xQKi31), I agree that the increase in computational cost seems somewhat excessive as it is greater than the prior method, TTDG.

---

> ### Author Response · Authors · 2025-11-27
> **Further Clarification**
>
> **1.Visual Evidence of the Separation Between Domain-Invariant and Domain-Specific Features**
>
> We have provided visual evidence demonstrating the separation between domain-invariant features and domain-specific features from individual domains. As shown in Figure 3 of the main manuscript, we fitted 600 features from three source domains with different distributions using two approaches: a single codebook strategy, and our multi-codebook strategy combining domain-specific and domain-invariant codebooks. We then visualized the source domain features preserved in the codebooks using t-SNE. (Existing research generally agrees that domain-invariant features across multiple domains tend to converge, while domain-specific features—which are key to distinguishing domains—typically exhibit distinct distributions.)
>
> (1) The single codebook strategy failed to effectively separate domain-specific features. As shown in Figure 3a the main manuscript, features from the three domains are compressed together, resulting in the loss of domain-specific characteristics and an inability to effectively distinguish between the domains.
>
> (2) In our combined strategy using domain-specific and domain-invariant codebooks, features represented by the domain-invariant codebook are shown in purple, while those represented by the three domain-specific codebooks are shown in red, green, and blue, respectively. Figure 3 of the main manuscript clearly demonstrates that the domain-invariant codebook effectively learned domain-invariant features, as features from all three domains exhibit similar distributions and are closely clustered. Meanwhile, the domain-specific features represented by the three domain-specific codebooks effectively distinguish between the domains, providing clear evidence that domain-specific features have been successfully disentangled.
>
> **2. Explanation Regarding Computational Efficiency**
>
> Existing Test-Time Domain Generalization (TTDG) methods are primarily designed for high-level vision tasks, where features represent abstract, image-level representations—essentially capturing the global style of an image. Therefore, these methods adapt the target domain's feature distribution by adjusting its style to align with the source domain. However, Super-Resolution (SR) is a pixel-level prediction task that requires fine-grained adjustments to every pixel in the image. As a result, existing TTDG methods are unsuitable for SR tasks.
>
> (1) As shown in Figure 2 of the main manuscript, we visualized the adjusted target domain distributions using existing TTDG methods and our proposed MC-TTDG. The results demonstrate that existing TTDG methods fail to effectively transfer the target domain distribution, whereas MC-TTDG successfully transfers it to closely resemble the source domain distribution.
>
> (2) As shown in Table 1 of this rebuttal, existing TTDG methods provide almost no performance improvement for SR tasks. In contrast, the proposed MC-TTDG method achieves significant performance gains compared to these existing approaches.
>
> (3) As shown in Table 2 of this rebuttal, the computational cost increase of MC-TTDG over existing TTDG methods (e.g., TTMC and TTDG ) is not substantial. Moreover, the additional computational cost is justified, as our method achieves pixel-level feature transformation while existing methods only perform style transformation.
>
> (4) As MC-TTDG represents the first investigation into test-time domain generalization for low-level vision tasks like SR, our primary objective was to achieve pixel-level feature transformation for target domains. We acknowledge that developing lightweight TTDG methods presents an important and valuable direction for future research.
>
> (5) This situation is analogous to the comparison between Transformer and CNN: despite Transformers having higher computational demands, their unique large receptive field characteristics have led to widespread adoption. Similarly, while Diffusion models require longer inference times than GANs, their high stability and diversity have also enabled broad application. Compared to existing TTDG methods, the proposed MC-TTDG achieves pixel-level feature transformation, which is indispensable for low-level vision tasks.
>
> > Table 1: Performance comparison with other TTDG methods
> >|Method|Pan PSNR↑|Pan SSIM↑|Pan LPIPS↓|Sony PSNR↑|Sony SSIM↑|Sony LPIPS↓|DSC PSNR↑|DSC SSIM↑|DSC LPIPS↓|
> >|-|-|-|-|-|-|-|-|-|-|
> >|DG-PIC|29.71|0.8135|0.4130|30.06|0.8305|0.4455|29.88|0.8470|0.4463|
> >|TTMC|30.26|0.8411|0.4523|30.56|0.8541|0.4069|30.70|0.8776|0.4128|
> >|TTDG|29.71|0.8135|0.4463|30.06|0.8305|0.4455|29.88|0.8470|0.4130|
> >|MC-TTDG (Ours)|31.15|0.8594|0.3593|31.29|0.8791|0.3157|31.21|0.8836|0.3583|
>
> >Table 2: Computational efficiency comparison with TTDG methods.
> >|Method|GFlops|Paras(M)|Memory(MB)|Infer times(s)|
> >|-|-|-|-|-|
> >|DG-PIC|68.812|97.86|915|0.063|
> >|TTMC|91.633|186.42|1276|0.084|
> >|TTDG|81.562|166.42|1078|0.074|
> >|MC-TTDG (Ours)|85.433|178.64|1176|0.081|

---

### Author Response · Authors · 2025-12-01
**Summary of Review Comments and Rebuttal [1/2]**

Our work has been positively acknowledged by all reviewers:

* Reviewer rKtr noted: "the paper is the first to …The proposed solution ... is an elegant and highly intuitive answer ... extensive validation bolsters the claim that the method is effective and generalizable. "
* Reviewer xQKi commented: "1. The writing is clear and easy to understand, and the methods are easy to implement. 2.The generalization ability on various SR test datasets is impressive."
* Reviewer Kv5y highlighted: " MC-TTDG is the first to adapt codebook-based representation learning to TTDG for low-level vision...The experimental design is comprehensive…"
* Reviewer iQVi emphasized: "…overcoming the limitations of previous style transfer approaches…A first test-time domain generalization method designed for low-level vision tasks."

**Initial Ratings and Changes**

Initial ratings: rKtr: 4, xQKi: 6, Kv5y: 4, iQVi: 6

Revised ratings: rKtr: 4, xQVi: 6, Kv5y: 6, iQVi: 6

1. **Before the OpenReview API bug occurred, Reviewer Kv5y confirmed that we had fully addressed all of their concerns. As a result, the reviewer raised the rating from 4 to 6.**

2. Before the OpenReview API bug occurred, reviewer rKtr acknowledged that our rebuttal had adequately addressed their initial concerns. Subsequently, they raised two new questions, indicating that resolving these would likely lead to a higher rating.
Due to the constraints imposed by the OpenReview API bug, reviewer rKtr was unable to respond to our further rebuttal. However, we believe we have effectively addressed their new concerns.

---

> ### Author Response · Authors · 2025-12-01
> **Summary of Review Comments and Rebuttal [2/2]**
>
> Below, we summarize the reviewers' comments and our corresponding rebuttals. (For more detailed information, please refer to the specific discussion with each individual reviewer.)
>
> # rKtr
> Initial rating: 4
>
> Before the OpenReview API bug occurred, reviewer rKtr acknowledged that our rebuttal had adequately addressed their initial concerns. Subsequently, they raised two new questions, indicating that resolving these would likely lead to a higher rating.
> Due to the constraints imposed by the OpenReview API bug, reviewer rKtr was unable to respond to our further rebuttal. However, we believe we have effectively addressed their new concerns.
>
> A summary of the new concerns and our corresponding rebuttals is provided below:
>
> |Questions & Concerns|Rebuttal|
> |-|-|
> |Providing evidence that the proposed method successfully disentangles domain-invariant features from domain-specific features| As shown in Figure 3 of the main manuscript, we provide visualizations that demonstrate our method's effectiveness in clearly separating domain-invariant features from domain-specific features.|
> |Justification for the increased computational overhead|(1) We have compared the proposed method with existing test-time domain generalization approaches in terms of both computational efficiency and performance. (2) While existing methods perform global style adjustment through mean and variance modification on features, they are fundamentally unsuitable for low-level vision tasks that require pixel-wise prediction. In contrast, our method achieves pixel-level feature transfer, significantly enhancing performance in low-level vision networks (As shown in Figure 2, Table 4 and Table 12 of the manuscript). This demonstrated performance improvement justifies the additional computational cost.|
>
>
>
>
>
> # xQKi
> Initial rating: 6
>
> Before the OpenReview API bug occurred, reviewer xQKi acknowledged that our rebuttal had adequately addressed their initial concerns. They subsequently raised two new questions.
> Similarly, due to constraints imposed by the OpenReview API bug, reviewer xQKi was unable to respond to our further rebuttal. However, we are confident that our responses have effectively addressed their new concerns.
>
> A summary of the new concerns and our corresponding rebuttals is provided below:
>
> |Questions & Concerns|Rebuttal|
> |-|-|
> |Comparison with fine-tuning methods| We have included comparisons with both fine-tuning methods and test-time domain adaptation methods.|
> |What are the advantages of this codebook-based approach?| We compared our method with existing test-time domain generalization methods in terms of performance and efficiency. The proposed method effectively achieves pixel-level feature transformation without requiring training on test samples, making it particularly suitable for pixel-wise prediction tasks such as image super-resolution.|
>
>
> # Kv5y
> Initial rating: 4
>
> **Before the OpenReview API bug occurred, Reviewer Kv5y confirmed that we had fully addressed all of their concerns. As a result, the reviewer raised the rating from 4 to 6.**
> Therefore, we have not provided a detailed summary of the questions and responses related to this reviewer.
>
>
> # iQVi
> Initial rating: 6
>
> |Questions & Concerns|Rebuttal|
> |-|-|
> |Suggestion to report the computational efficiency of the proposed method| We have added comparisons of computational efficiency between the proposed method and existing test-time domain generalization methods. Additionally, we have included comparisons with both fine-tuning methods and test-time domain adaptation methods, which also utilize test samples to enhance network performance.|
> |Does the number of codebooks affect network performance?| We have supplemented the ablation study investigating the impact of varying the number of codebooks on network performance.|
> |In a multi-codebook setting, do issues like codebook redundancy or imbalance exist, and how are they mitigated?|(1) We have included additional ablation experiments analyzing the effect of different codeword quantities on network performance. (2) We have discussed the diversity regularization loss employed to address the issue of codeword redundancy.|
> |Can the method's performance be validated on datasets from unknown cameras?|In Table 13 of the main manuscript, we have demonstrated the effectiveness of the proposed method on several datasets featuring unknown cameras, including B100, Manga109, Set14, DIV2K, Urban100, and Set5.|

---

### Meta-Review · Area_Chair_QAkE · 2025-12-24

**Summary:**

This paper studies test-time domain generalization for image super-resolution and proposes a multi-codebook based framework (MC-TTDG) to enable pixel-level feature transfer at inference time. Reviewers generally agree that the problem setting is well motivated and that existing style-transfer-based TTDG methods are indeed ill-suited for low-level, pixel-wise prediction tasks. The proposed use of domain-invariant and domain-specific codebooks, together with a voting-based selection strategy, is considered intuitive and practically effective, and the experimental results consistently demonstrate strong performance gains across datasets and architectures.

After considering the rebuttal and post-discussion feedback, some concerns remain regarding the conceptual novelty of the disentanglement mechanism and the lack of fully principled or quantitative validation of domain-invariant versus domain-specific feature separation. Nevertheless, reviewers acknowledge that the method is carefully engineered, empirically solid, and represents one of the first systematic attempts to adapt TTDG to low-level vision tasks such as super-resolution. While the work may not yet offer a fundamentally new theoretical formulation of disentanglement, its clear empirical benefits, novel problem focus, and potential to stimulate further research in this underexplored setting support acceptance as a poster. The authors are suggested to open-source the proposed method upon acceptance.

**Reviewer Concerns:**

Reviewer rKtr: While the approach is intuitive and effective, concerns remain about whether performance gains alone sufficiently validate the claimed disentanglement between domain-invariant and domain-specific features.

Reviewer xQKi: The method is clearly presented and empirically strong, but comparisons with full fine-tuning and the efficiency–performance trade-off could be further clarified.

Reviewer Kv5y: Most technical concerns were addressed in the rebuttal, and the reviewer views the method as a meaningful and well-validated extension of TTDG to super-resolution.

Reviewer iQVi: The idea is interesting and practically valuable at a borderline acceptance level, and the reviewer encourages acceptance and open-sourcing to facilitate further progress in the community.

**Reviewer Scores:**

Reviewer rKtr: Likely no change.

Reviewer xQKi: Likely no change.

Reviewer Kv5y: Increased to a positive score after rebuttal.

Reviewer iQVi: Likely no change.

---

### Decision · Program_Chairs · 2026-01-26

Accept (Poster)